

# Near-surface and path-averaged mixing ratios of NO$_2$ derived from car DOAS zenith-sky and tower DOAS off-axis measurements in Vienna: a case study

**Stefan F. Schreier[1], Andreas Richter[2], and John P. Burrows[2]**

[1]Institute of Meteorology, University of Natural Resources and Life Sciences, Vienna, Austria

[2]Institute of Environmental Physics, University of Bremen, Germany

Correspondence to: S. F. Schreier (stefan.schreier@boku.ac.at)

**Abstract.** Nitrogen dioxide (NO$_2$), produced as a result of fossil fuel combustion, biomass burning, lightning, and soil emissions, is a key urban and rural tropospheric pollutant. In this case study, ground-based remote sensing has been coupled with the in situ network in Vienna, Austria, to investigate NO$_2$ distributions in the planetary boundary layer. Near-surface and path-averaged NO$_2$ mixing ratios within the metropolitan area of Vienna are estimated from car DOAS (Differential Optical Absorption Spectroscopy) zenith-sky and tower DOAS horizon observations. The latter configuration is innovative in the sense that it obtains horizontal measurements at more than hundred different azimuthal angles – within a 360° rotation taking less than half an hour. Spectral measurements were made with a DOAS instrument on nine days in April, September, October, and November 2015 in the zenith-sky mode and on five days in April and May 2016 in the off-axis mode. The analysis of tropospheric NO$_2$ columns from the car measurements and O$_4$ normalized NO$_2$ path averages from the tower observations provide interesting insights into the spatial and temporal NO$_2$ distribution over Vienna. Integrated column amounts of NO$_2$ from both DOAS-type measurements are converted into mixing ratios by different methods. The estimation of near-surface NO$_2$ mixing ratios from car DOAS tropospheric NO$_2$ vertical columns is based on a linear regression analysis including mixing-height and other meteorological parameters that affect the dilution and reactivity in the planetary boundary layer – a new approach for such conversion. Path-





averaged NO₂ mixing ratios are calculated from tower DOAS NO₂ slant column densities by taking
into account topography and geometry. Overall, lap averages of near-surface NO₂ mixing ratios
obtained from car DOAS zenith-sky measurements, around a circuit in Vienna, are in the range of
3.8 to 26.2 ppb and in good agreement with values obtained from in situ NO₂ measurements for
days with wind from the Southeast. Path-averaged NO₂ mixing ratios at 160 m above the ground
as derived from the tower DOAS measurements are between 2.5 and 9 ppb on two selected days
with different wind conditions and pollution levels and show similar spatial distribution as seen in
the car DOAS zenith-sky observations. We conclude that the application of the two methods to
obtain near-surface and path-averaged NO₂ mixing ratios is promising for this case study.
**1 Introduction**
Tropospheric nitrogen oxides (NOₓ = NO + NO₂) are released from various human activities and
natural sources (Lee et al., 1997). Fossil fuel combustion to produce energy results in NOₓ
emissions by traffic, industry and domestic heating or cooling appliances. Nitric oxide (NO) is the
predominant part of NOₓ emitted from these sources. However it is rapidly converted to nitrogen
dioxide (NO₂) by reaction with ozone (O₃). During daytime, given sufficient ultraviolet radiation,
NO₂ is photolysed to produce NO and oxygen atoms. The reaction of oxygen atoms with molecular
oxygen (O₂) results in the production of O₃. Under polluted conditions the so called Leighton
photostationary state is established. However, as the NOₓ air is mixed in daylight with
hydrocarbons and being diluted, the catalytic production of O₃ results and nitric acid (HNO₃) is
formed. The latter is absorbed on aerosols, which are also produced in air masses generating
photochemical smog.
Although NOₓ concentrations are relatively low in the atmosphere, these reactive gases play a
significant role in atmospheric chemistry, air pollution, and climate change, in particular in urban
environments (e.g. WHO, 2003; IPCC, 2013). For example, elevated levels of air pollutants such
as NO₂ and O₃ affect human health (e.g. Dockery et al., 1993), as the long-term exposure to these
gases can influence mortality and morbidity (e.g. Künzli et al., 2000).





In addition to the in situ $NO_2$ measurement techniques such as chemiluminescence monitors (e.g. Fontijn et al., 1970), the differential optical absorption spectroscopy (DOAS) method (Perner and Platt, 1979) can also be used to quantify atmospheric $NO_2$ concentrations. Nowadays, the DOAS technique is a widely-used remote sensing method to retrieve the amount of several trace gases having narrow band absorption structures in the UV and visible part of the electromagnetic spectrum. The (passive) DOAS principle, which is based on Lambert-Beer's law, can be applied to measurements from various ground-based, ship-based, aircraft-based, and satellite-based platforms (e.g. Platt and Stutz, 2008 and references therein).

The great advantage of satellite-based measurements is their daily (near) global coverage and thus, the possibility to evaluate temporal trends above selected regions. However, it is difficult to resolve $NO_2$ at the city scale because of the coarse resolution of satellite sensors (Richter et al., 2005; Hilboll et al., 2013). Aircraft-based measurements deliver higher resolved images of the spatial $NO_2$ distribution along a given flight track, but only during short-term measurement campaigns (Heue et al., 2005; Wang et al., 2005; Schönhardt et al., 2015; Meier et al., 2017; Nowlan et al., 2018). As is the case for aircraft-based DOAS measurements of $NO_2$, ship-based observations of $NO_2$ are also usually performed on a campaign basis (Peters et al., 2012; Takashima et al., 2012; Schreier et al., 2015; Hong et al., 2018). Finally, information on tropospheric $NO_2$ can also be obtained from ground-based platforms using the Multi AXis (MAX) DOAS system (Hönninger et al., 2004; Wittrock et al., 2004). In contrast to other platforms, ground-based DOAS measurements are usually performed continuously and at fixed locations.

More recently, DOAS-type measurements of $NO_2$ are also performed from a car, which enables the observation of the horizontal variation of tropospheric $NO_2$, in addition to its temporal evolution. Such observations have been used for the quantification of total emissions from cities and/or known emission sources (Johansson et al., 2008; Rivera et al., 2009; Ibrahim et al., 2010; Shaiganfar et al., 2011; Wang et al., 2012; Frins et al., 2014; Ionov et al., 2015), for the estimation of emission fluxes from cities (Johansson et al., 2009; Rivera et al., 2013), for the comparison with satellite observations of $NO_2$ (Wagner et al., 2010; Constantin et al., 2013; Wu et al., 2013), for the comparison with model simulations (Dragomir et al., 2015), and for the validation of airborne measurements of $NO_2$ (Meier et al., 2017; Tack et al., 2017; Merlaud et al., 2018). While some of the mentioned studies use the MAX-DOAS measurement principle, others apply their instruments





in the zenith-sky viewing mode only. The main challenges for the retrieval of tropospheric $NO_2$ for
the latter approach are obtaining accurate knowledge of the $NO_2$ signal in the reference
measurement as well as the removal of the stratospheric $NO_2$ contribution (as a function of SZA).
Both quantities cannot directly be separated from the zenith-sky measurements alone and thus, not
accounting for these contributions can lead to large errors, especially in regions with low $NO_2$
levels (Wagner et al. 2010). Therefore, approaches were developed to estimate these contributions
by using additional data and methods. While the stratospheric $NO_2$ amounts can be obtained from
satellite measurements in combination with atmospheric modelling, the background signal in the
reference spectrum can be estimated by calculating a reference measurement applying the Langley-
plot method (Constantin et al. 2013). Another approach to estimate the background signal in the
reference spectrum would be to utilize $NO_2$ concentration measurements from nearby in situ
monitoring stations and convert those quantities into tropospheric $NO_2$ vertical columns, e.g. by
applying an empirical relationship (Kramer et al., 2008).
The aims of the present study are two-fold. Firstly, it attempts to build on earlier work and
investigates the spatial and temporal variability of $NO_2$ pollution in Vienna by using a simple
zenith-sky telescope and a miniature spectrometer operated from a normal car. The relatively large
number of air quality monitoring stations in and around Vienna, including continuous
measurements of $NO_2$ concentrations at surface level, provides the prerequisites for a comparison
between these two observation systems which has not yet been performed in past studies. Secondly,
the potential of DOAS horizon measurements, performed with the same instrument on a rotating
tower platform in Vienna is investigated – a DOAS-type approach to gain detailed horizontal $NO_2$
distributions on the city-scale within less than half an hour. Our tower DOAS off-axis observations
can be best compared to the measurement configuration of the CU 2-D-MAX-DOAS instrument
during the Multi-Axis DOAS Comparison campaign for Aerosols and Trace gases (MAD-CAT) in
Mainz, Germany (Ortega et al., 2015). The authors of that study developed a four-step retrieval to
derive, amongst other parameters, near-surface horizontal distributions of $NO_2$ at 14 pre-set
azimuth angles distributed over a 360° view. The tower DOAS off-axis configuration presented in
our study is in the sense innovative that it is the first approach having more than 100 horizontal
measurements within a 360° rotation that lasts less than half an hour. Also new is the performance
of the DOAS instrument at an altitude of more than 100 meters above ground, which gives insights



into the vertical variability of $NO_2$ within the planetary boundary layer over the urban environment
of Vienna, when these measurements are combined with ground-based in situ data. The horizontal
optical path lengths in our study are estimated by making use of the combination of geometry and
topography. We note that the discussion of tower DOAS off-axis measurements is only based on a
couple of data available. Further measurements on a routine base could serve as a data set to go
more in detail and estimate the 3-D distribution of trace gases, as shown in Ortega et al. (2015).
From both DOAS-type columnar $NO_2$ measurements reported in our study, near-surface and path-
averaged $NO_2$ mixing ratios are estimated by using both existing methods and a novel linear
regression analysis. These measurements provide insights about the $NO_2$ distributions in the
Viennese boundary layer, which are interestingly in themselves but could also help in deciding
where to place an optimal set of MAX-DOAS instruments around the capital and largest city of
Austria. The proposed long-term measurements of these instruments, which are foreseen in the
VINDOBONA (VIenna horizontal aNd vertical Distribution OBservations Of Nitrogen dioxide
and Aerosols) project (www.doas-vindobona.at), will provide a valuable data set for analyzing the
temporal variability of air pollutants over Vienna.
The city of Vienna has the second largest number of inhabitants (about 1.8 million) within German
speaking countries. It is part of a metropolitan area having a population of 2.8 million and is a
typical example of a growing city (www.statistik.at). There are many $NO_x$ emission sources such
as high-traffic roads, individual power plants, and industrial buildings that contribute to increased
levels of $NO_2$. The Environment Agency Austria reported a significant decrease in $NO_x$ emissions
from traffic and industry since 2005 in Austria, which is mainly because of the progress in
automotive technology. However, they also highlighted the fact that a defined legal limit of annual
mean $NO_2$ concentrations (35 µg/m³) was still exceeded in the past years at several Austrian air
quality monitoring stations – including stations in Vienna (Spangl and Nagl, 2016). In the year
2015, annual mean $NO_2$ concentrations exceeded the legal limit at one station in Vienna. Moreover,
hourly limit values (200 µg/m³) were exceeded several times at four stations. We note that $NO_2$
levels didn't exceed the legal limits on the days of measurements presented in this case study.
However, a substantial number of hourly values with $NO_2$ concentrations higher than 100 µg/m³
were observed on these days.



In the following Sect. 2, the DOAS instrument and the setups for the car DOAS zenith-sky and tower DOAS off-axis measurements are introduced. Details about the data analysis, including the retrieval of columnar tropospheric $NO_2$ amounts and the conversion into mixing ratios are given in Sect. 3. The results of this study are described and discussed in Sect. 4, followed by a short summary and outlook (Sect. 5).

## 2 Instrument and car journeys

### 2.1 DOAS instrument

For the car DOAS zenith-sky and tower DOAS off-axis observations of tropospheric $NO_2$ in Vienna, a DOAS system was used to measure scattered sunlight from directly overhead and from the horizon, respectively. A cardboard box was built to house a commercial Avantes miniature spectrometer (AvaSpec-ULS2048x64) and a notebook. The AvaSpec-ULS2048x64 is small in size (175 x 110 x 44 mm), robust, and lightweight (855 grams). The instrument performs spectral measurements between 290 and 550 nm at a spectral resolution of 0.65 nm. Both the spectrometer and notebook where supplied with electricity from the car battery and from the existing tower power circuit during the measurements.

### 2.2 Setup of the car DOAS zenith-sky measurements

An optical fibre was connected to the spectrometer and threaded through an aluminium bracket to the outside of the car, where it was fixed to a small aluminium plate by duct tape. In order to prevent direct sunlight from entering the optical fibre, a cylindrical plastic tube was used for shading the entrance. The field of view of the optical fibre was characterized in the laboratory to be about ±5°. As the telescope was directed to the zenith, no large errors are expected for the retrieval of tropospheric vertical $NO_2$ columns in this case as light path length is relatively insensitive to small deviations of the pointing from the zenith direction. For stability reasons, the bracket was clamped by the two door windows of the rear area. The geographical position of the car was recorded by a GPS receiver, which was connected to and powered by a USB port of the laptop computer.





The overall approach was to keep the measurement system simple. Therefore, only the zenith-
direction was implemented which is insensitive to changes in pointing as from telescope
misalignments or car movements. Pointing the instrument closer to the horizon increases the
sensitivity to tropospheric $NO_2$, but introduces additional complication as pointing accuracy in a
moving car becomes an issue. Experience also showed that in a city environment, a large fraction
of the measurements at 22° or 30° elevation, for example, is affected by blocking from houses,
trees or other vehicles. As shown in previous studies (e.g. Wagner et al., 2010; Shaiganfar et al.,
2011), the air mass factor for measurements at 22° and 30° depends on the relative azimuth between
the telescope orientation and the sun, necessitating computation of the car heading from GPS data
which can be complex in typical city traffic situations. In summary, the choice was made to use a
simple and robust method at the expense of reduced sensitivity.
A total of twenty identical car circuits around Vienna were performed on nine days in April,
September, October, and November 2015 within the metropolitan area of Vienna (see Table 2).
Each drive spanned about 110 km and lasted about 1.5 hours. In order to minimize the effect of
clouds and wind speed, measurements were performed in the morning rather than in the afternoon.
After a successful test phase of the car DOAS zenith-sky measurements on 10 April 2015, more
days were planned in fall of the same year, including working days and days on weekends as well
as days with different wind conditions. Measurements between April and September, e.g. during
the summer season, were unfortunately not possible due to other priorities and due the fact that the
authors were not located in Vienna during that time.
Figure 1 illustrates an exemplary overview of a single car journey performed on 10 April 2015
between 5:27 and 6:59 UT. The starting point of each drive was within the Municipality of
Wolkersdorf im Weinviertel (48° 22' 59'' N, 16° 31' 05'' E), a small city located in Lower Austria,
about 10 km north of Vienna and away from large sources of $NO_x$. From there, the journey was
planned to cover one of the busiest motorways in Austria, pass by known emission sources (e.g.
power plants), and drive round one of the largest inland refineries in Europe, before heading back
to the starting point on a different route.
**2.3 Setup of tower DOAS off-axis measurements**



The same DOAS instrument was used for the measurements performed from the Café at the Vienna
Danube Tower (48° 14′ 25″ N, 16° 24′ 36″ E), which is rotating at about 160 m above ground
(www.donauturm.at). Due to its geographical location (about 4.5 km to the northeast of the city
center), it is possible to scan both urban and rural areas during a single anti-clockwise 360° rotation
(duration = 26.5 minutes). In contrast to the car DOAS zenith-sky measurements (see Sect. 2.2),
the telescope was directed towards the horizon at an elevation angle of 0°. An optical lens was
placed in front of the light fibre entrance to reduce the field of view of the instrument to about 0.8°.
Both the lens and the entrance of the optical fibre were protected from direct sunlight by a purpose-
built cardboard. Because the scattered sunlight was passing through a thick glass window, no UV
spectra could be recorded by the DOAS instrument from the rotating tower platform.
Tower DOAS off-axis observations were performed on five days in April and May 2016. More
than thirty 360° scans of Vienna were recorded, each of them for an individual rotations of the
Cafe. For reasons of simplicity and accessibility, zenith-sky measurements were only taken
afterwards from the open terrace, which is located a few meters below.
As the Vienna Danube Tower does not provide information on the exact orientation of the platform,
and due to the fact that the signal of the GPS receiver was not accurate enough to reliably determine
the position along the circle, the horizontal viewing angle was determined by the following
approach: The DC Tower 1, the tallest skyscraper in Austria, which is located in about 1 km
distance from the Vienna Danube Tower, comes into field of view once every rotation and
considerably reduces the signal (see Fig. 2). According to Google Earth, the position of the Vienna
Danube Tower relative to this skyscraper is 167° (nearly south). By assuming that the rotation
speed and the alignment on intensity minima are constant, the horizontal viewing angle can be
determined from the periodic sharp reduction in intensity.
**3 Data analysis**
**3.1 DOAS analysis**
The spectral measurements as obtained during the individual car journeys and tower platform
rotations are analyzed using the DOAS technique applying a nonlinear least-squares fitting


algorithm. The spectral retrieval of $NO_2$ differential slant column densities (DSCDs) is based on a
fitting window between 425 and 490 nm, a polynomial degree of five (car DOAS zenith-sky) and
seven (tower DOAS off-axis), and a wavelength calibration using data from the Solar atlas of
Kurucz et al., (1984). These general settings have been commonly used in recent studies for the
retrieval of $NO_2$ DSCDs from ground-based DOAS-type measurements (e.g. Roscoe et al., 2010).
High resolution absorption cross-sections of $O_3$, $NO_2$, $O_4$, $H_2O$, and a pseudo-cross section
accounting for rotational Raman scattering as computed with QDOAS (Danckert et al., 2015) have
been included in the two retrieval settings (see Table 1). The motivation for using a higher
polynomial degree in the analysis of the horizon measurements are large broadband residuals found
in the data. These residuals are attributed to the fact that the horizon measurements were taken
through thick multi-layer glass while the zenith-sky measurement was taken outdoors.
Exemplary car DOAS zenith-sky fit results, recorded on 10 April 2015 (SZA = 47.68°) under
elevated $NO_2$ pollution (DSCD = 4.02 x $10^{16}$ molec cm$^{-2}$), are shown in Fig. 3 (left panels). In some
parts of the route, the zenith view of the instrument is obstructed by tunnels, bridges or other
objects. These measurements were identified using an intensity criterion and removed from the
data set. However, some outliers having unrealistically high values of $NO_2$ are still present in the
data set, which strongly correlate with exceptional high chi-square values. Consequently, we only
consider $NO_2$ DSCDs with chi-square values < 2.5 x $10^{-3}$ for further analysis. Noontime
measurements of three selected days, taken in rural areas and close to air quality monitoring stations
are used as reference measurements (see Sect. 3.2.3 and Table 2).
Exemplary tower DOAS off-axis fit results obtained from a spectra recorded on 29 April 2016
(SZA = 66.99°) under elevated $NO_2$ pollution (DSCD = 1.46 x $10^{17}$ molec cm$^{-2}$) are shown in Fig.
3 (right panels). When comparing the two fit results, it becomes clear that the absorption by $NO_2$
in the horizontal path is larger by a factor of 3.6 in this case. This is because most of the $NO_2$ in
urban environments is found in the boundary layer, close to the ground.
**3.2 Car DOAS measurements of tropospheric NO$_2$**
**3.2.1 Temporal resolution and computation of horizontal NO$_2$ gradients**



In order to obtain some information about the signal to noise ratio of the instrument and the
horizontal gradients of $NO_2$ present in the city, the temporal resolution of the car DOAS zenith-
sky measurements was initially set to 0.5 seconds. The collected spectra were then averaged over
intervals of 5 seconds (see Fig. 4), which corresponds to a traveled distance of about 100 m. An
averaging interval of 5 seconds was also used by Constantin et al. (2013) for their mobile
measurements.
The upper panel in Fig. 4 shows the temporal evolution of $NO_2$ DSCDs on 3 November 2015. The
red and blue lines represent the full resolution of 0.5 seconds and averaged values, respectively.
While the full resolution is noisy (maximum deviation ~ 5 x $10^{15}$ molec cm$^{-2}$), the averaged values
follow the general pattern of $NO_2$ along the car drives. For better clarity, the middle panel illustrates
a shorter section of that day, indicated by the green lines in the upper panel. The same is true for
the lower panel, which represents a short section of the middle panel. Based on these results we
argue that the selection of 5 seconds as an averaging interval appears to be optimal and a good
compromise in our study, in spite of more information being found in the high-frequency data in
some cases.
In addition to mapping the spatial distribution of $NO_2$ in Vienna, it is also interesting to evaluate
typical horizontal gradients within the city. The identification of such mean horizontal gradients of
$NO_2$ along the individual car routes is based on the following approach. Firstly, horizontal distances
between start and end point of individual car DOAS zenith-sky measurements at the full resolution
of 0.5 seconds are calculated and summed. Secondly, $NO_2$ DSCDs at the same time resolution are
interpolated on 100 m bins as obtained from the first calculation step. Thirdly, absolute differences
of $NO_2$ DSCDs are derived for each pair of consecutive interpolated values within 5 km. In a final
step, absolute differences are averaged along the car track in order to compute a mean horizontal
gradient for each single car lap.
**3.2.2 Stratospheric $NO_2$ columns**
The stratospheric correction in our study is based on stratospheric $NO_2$ fields as simulated by the
Bremen 3d CTM (B3dCTM) and scaled to satellite observations from the Global Monitoring



Instrument 2 (GOME-2) over a selected region in the Pacific (180°-140° W, 48°-48.5° N). This scaling is necessary as there is an offset between modeled and measured $NO_2$ amounts.

Briefly, the B3dCTM, which evolved from SLIMCAT (Chipperfield, 1999), is a combined model approach based on the "Bremen transport model" (Sinnhuber at al., 2003a) and the chemistry code of the "Bremen two-dimensional model of the stratosphere and mesosphere" (Sinnhuber et al., 2003b; Winkler et al., 2008). It is driven by ECMWF ERA Interim meteorological reanalysis fields (Dee et al., 2011).

Exemplary simulated stratospheric $NO_2$ columns above Vienna as obtained from B3dCTM are shown in Fig. 5 for 19 October 2015. While stratospheric $NO_2$ amounts sharply decrease in the morning due to photolysis of $NO_2$, the observed increase of $NO_2$ over the day is the result of dinitrogen pentoxide ($N_2O_5$) photolysis. The green rectangle indicates the start (06:57 UT) and end time (09:56 UT) of car DOAS zenith-sky measurements performed on 19 October 2015 (see also Table 2).

### 3.2.3 Conversion to tropospheric $NO_2$ vertical column densities

The conversion of $NO_2$ DSCDs obtained from car DOAS zenith-sky measurements into $NO_2$ tropospheric vertical column densities ($VCD_{tropo}$) is based on the approach by Wagner et al. (2010) and Constantin et al. (2013). The authors of the latter study have used a similar zenith-sky DOAS system on a car to derive tropospheric $NO_2$ amounts in Romania. $VCD_{tropo}$ from car DOAS zenith-sky measurements is determined via the following equation:

$$VCD_{tropo} = \frac{DSCD_{meas} + SCD_{ref} - VCD_{strato} * AMF_{strato}}{AMF_{tropo}}, \tag{1}$$

where $DSCD_{meas}$ is obtained from the car DOAS zenith-sky measurements by applying the DOAS analysis (see Sect. 3.1). $SCD_{ref}$ is the slant column in the reference spectrum, which cannot be measured directly when applying the zenith-sky viewing mode only. Moreover, $SCD_{ref}$ has both stratospheric and tropospheric amounts, which are estimated with different approaches in the literature. The tropospheric $NO_2$ signal in $SCD_{ref}$ in our study is calculated by applying the empirical relationship between $VCD_{tropo}$ and in situ $NO_2$ mixing ratios as reported in Kramer et al.





(2008). To be more specific, the estimation of tropospheric $NO_2$ amounts in $SCD_{ref}$ is conducted
for the time and location of the three selected reference measurements taken in rural areas outside
the boundaries of Vienna and about 13 km (10 April 2015) and 3 km (27 September and 23
October) away from the nearest air quality monitoring station. More details on data from air quality
monitoring stations are given in the following Sect. 3.3. $VCD_{strato}$ is derived from B3dCTM
simulations and scaled to GOME-2 observations (see Sect. 3.2.2). Stratospheric and tropospheric
airmass factors are calculated with the SCIATRAN radiative transfer model (Rozanov et al., 2014).
For the latter case, the AMF is calculated for a wavelength of 460 nm by assuming that $NO_2$ is
well-mixed between the ground surface and an altitude of 1 km. The computed stratospheric and
tropospheric AMFs as a function of SZA are shown in the left and right panels of Fig. 6,
respectively.
**3.3 In situ measurements of $NO_2$**
For the estimation of tropospheric $NO_2$ amounts in $SCD_{ref}$ as well as for the comparison of $NO_2$
$VCD_{tropo}$ obtained from car DOAS zenith-sky measurements with in situ $NO_2$ concentrations, data
from more than a dozen air quality monitoring stations in and around Vienna, provided by the
Environment Agency Austria, UBA (Umweltbundesamt), are used.
Tropospheric $NO_2$ amounts in $SCD_{ref}$ are calculated by converting simultaneous in situ $NO_2$
measurements from the air quality monitoring stations in Gänserndorf (10 April 2015) and
Wolkersdorf (27 September and 23 October 2015) into $VCD_{tropo}$ applying the empirical
relationship between concurrent MAX-DOAS and urban background in situ measurements
(Kramer et al., 2008):
$$y = 0.036x + 0.018, \qquad\qquad (2)$$
where y is the tropospheric $NO_2$ VCD (tropospheric $NO_2$ amount in $SCD_{ref}$ in our study) in units
of $10^{16}$ molec $cm^{-2}$ and x denotes the in situ $NO_2$ mixing ratios in units ppb. The conversion of in
situ $NO_2$ concentrations into in situ mixing ratios in our study is described in Sect. 3.5.





The tropospheric background values in $SCD_{ref}$ as determined with Eq. 2 are estimated at $1.3 \times 10^{15}$,
$1.1 \times 10^{15}$, and $2.2 \times 10^{15}$ molecules cm$^{-2}$ on 10 April, 27 September, and 23 October, respectively.
We note that the extrapolation of the empirical relationship to our measurements is critical in a
sense that meteorological conditions and emissions are not the same in Leicester and Vienna. Due
to the fact that $SCD_{ref}$ measurements were taken outside of Vienna in our study, with in situ
measurements of $NO_2$ being in the range of 2.5 to 6 ppb on those three days and indicating rather
low tropospheric $NO_2$ amounts, the error is assumed to be likewise low in this case.
For the comparison of car DOAS zenith-sky and in situ $NO_2$ observations, we have selected half-
hour averages of $NO_2$ concentrations from seven stations in Lower Austria and eight stations in
Vienna that are within 5 km from the car route (see Table 3). For both cases, half-hour averages of
$NO_2$ concentrations are converted into mixing ratios (see Sect. 3.5).
**3.4 Mixing-height from ceilometer observations**
The conversion of $VCD_{tropo}$ into mixing ratios as described in the following Sect. 3.5 requires,
besides meteorological measurements of pressure and temperature, information on the planetary
boundary layer depth (also known as mixing-height). The Austrian official weather service, ZAMG
(Zentralanstalt für Meteorologie und Geodynamik), performs operational aerosol-layer height
measurements with a Vaisala CL51 ceilometer at the Hohe Warte site in the North West of Vienna
(48° 14' 55'' N, 16° 21' 23'' E). Mixing-height (MH) time series are obtained from these
measurements by removing unrealistic nocturnal aerosol-layer height values, avoiding outliers,
filling data gaps by linear interpolation, and smoothing (Lotteraner and Piringer, 2016). Mixing-
height data at a temporal resolution of 5 minutes were provided by ZAMG for those days when car
DOAS zenith-sky measurements were carried out.
**3.5 Comparison of $NO_2$ mixing ratios obtained from car DOAS zenith-sky and in situ**
**measurements**



The comparison between the two independent $NO_2$ observations (car DOAS zenith-sky versus in
situ) is based on gridding the data of both measurement techniques onto a 0.01° x 0.01° spatial
resolution. For a better comparison, $NO_2$ $VCD_{tropo}$ as obtained from car DOAS zenith-sky
measurements as well as in situ $NO_2$ concentrations are converted into mixing ratios. The former
conversion is based on recommendations made in Knepp et al. (2013). The authors of that study
have converted Pandora tropospheric $NO_2$ values into mixing ratio values by applying a planetary
boundary layer (PBL) height correction factor. Although this approach assumes a constant mixing
ratio in the PBL, which is not necessarily correct in an urban environment, it accounts for the
variability in MH throughout the day. We follow their approach and estimate boundary layer
mixing ratios of $NO_2$ ($X_{NO2}$) via the following equation:
$$Car\ DOAS\ (BL)\ X_{NO_2} = \frac{VCD_{tropo}}{MH * n_a}, \qquad\qquad (3)$$
where MH is the mixing-height (PBL in their study) and $n_a$ denotes the number density of air (N
in their study). Here, we use lap averages for MH as calculated from the data in 5 minute resolution
provided by ZAMG (see Sect. 3.4). The standard deviation of these lap averages generally ranges
between 10 and 50 m but can be as high as 200 m when wind speeds are high (see Table 2). The
number density of air, which is related to the atmospheric pressure by the ideal gas law, is also
averaged over the individual car laps. Meteorological measurements of pressure (p) and
temperature (T) used for the calculation of $n_a$ are provided by the BOKU (Universität für
Bodenkultur) weather station, located in the North West of Vienna (48° 14' 16.45" N, 16° 19' 54"
E). We note that the weather station is located about 100 m higher than the altitude level of the car
route. Thus, pressure might be slightly lower when compared to the pressure level 100 m below.
On the other hand, the weather station is also located outside of the city center, at the foot of the
hills in the Northwest and in a less densely populated residential area with many green areas,
resulting in slightly cooler temperatures than expected for other places along the car route.
Following this reasoning, it becomes clear that the altitude difference might cancel in the
calculation of $n_a$ (see Eq. 3) and also in the following Eq. 5.
Recently, Dieudonne et al. (2013) highlighted the fact that large vertical gradients of $NO_2$
concentrations exist over urban areas. The authors of that study suggest that the averaged
concentration within the PBL is only about 25% of $NO_2$ surface concentration measurements when



NO$_2$ profiles from chemistry-transport models are assumed for the PBL. Following this reasoning,
Car DOAS (BL) X$_{NO2}$ as estimated via Eq. 3 does not represent NO$_2$ near-surface mixing ratios
sufficiently well and a comparison with NO$_2$ as obtained from air quality monitoring stations and
converted into In situ X$_{NO2}$ (see Eq. 5) is not yet reasonable. Consequently, an empirical approach
for estimating near-surface NO$_2$ mixing ratios from the car DOAS zenith-sky measurements was
developed, in addition to Car DOAS (BL) X$_{NO2}$.
In order to achieve optimal agreement between car DOAS zenith-sky measurements and in situ
observations in our study, we include four parameters that are expected to affect the vertical NO$_2$
gradients and conduct a linear regression analysis as follows:
$$Y = \beta_0 + \beta_1 X_1 + \beta_2 X_2 + \beta_3 X_3 + \beta_4 X_4 + \varepsilon, \tag{4}$$
where $Y$ is the expected value of the dependent variable In situ X$_{NO2}$ and $X_1$, $X_2$, $X_3$, and $X_4$ are the
independent variables VCD$_{tropo}$ NO$_2$, MH, wind speed, and n$_a$, respectively (see Table 2).
The conversion of in situ NO$_2$ concentrations (c$_m$) into mixing ratios is based on the equation:
$$In\ situ\ X_{NO_2} = c_m \frac{1}{M_i} * \frac{RT}{p}, \tag{5}$$
where M$_i$ is the molecular weight of NO$_2$ and R denotes the universal gas constant. As for the
calculation of n$_a$, p and T measurements at a 10 minute resolution are taken from the BOKU
weather station and averaged for the individual car laps.
All NO$_2$ mixing ratio values within individual grid cells are averaged and then compared with each
other.
**3.6 Meteorological measurements of wind direction and wind speed**
Most of the emission sources other than traffic are located in the South-East of Vienna. The wind
blew exactly from this direction on several days when car DOAS zenith-sky measurements were
carried out. In addition, the car journey was planned to include the motorway along the Danube
River, spanning a distance of about 20 km from North-West (48° 21' 25'' N, 16° 18' 25'' E) to



South-East (48° 12' 32'' N, 16° 26' 24'' E). These are prerequisites for the optimal analysis of the
evolution of $NO_2$ in space and time, in particular on days where wind was blowing either from
North-West (NW) or South-East (SE). As there are no large sources of $NO_x$ located in the NW, we
rather focus on days when wind was blowing from the SE.
Data on wind direction and wind speed are provided by ZAMG. We have selected such data from
four stations in Lower Austria and five stations in Vienna that are in close proximity to the car
route (see Table 4). The temporal resolution of these measurements is 10 minutes. Instead of
attempting to map the wind direction to the car route in time, we have averaged these measurements
over the period between start and end time of each car journey and calculated the standard deviation
(see Table 2).
**3.7 Tower DOAS measurements of tropospheric $NO_2$**
**3.7.1 Temporal resolution and normalization of $NO_2$ DSCDs with $O_4$**
Compared to the car DOAS zenith-sky measurements, the temporal resolution of spectral
measurements performed on the rotating tower platform is higher (0.025 s). This is because of the
relatively fast rotation speed resulting in a full 360° rotation within only 26.5 minutes. Again, these
temporally high-resolved spectral measurements are averaged over 10 seconds. After the averaging
procedure, roughly 150 measurements remain for a single 360° rotation. These observations are
then interpolated on 3.6° segments, resulting in 100 measurements for one single rotation.
One of the main drawbacks of the measurements is that only one reference measurement was taken
after the measurements. This was because no zenith-sky measurement was possible from within
the restaurant, and no second DOAS system was available during that time for parallel
measurements from the surface. Therefore, a fixed zenith spectrum has to be used instead of a
sequential one, resulting in an increasing effect of a changing tropospheric light path (e.g. due to
geometry, aerosols, phase function etc.) with increasing time difference between the off-axis and
fixed zenith spectra. One way of overcoming this problem is to normalize $NO_2$ DSCDs with $O_4$
DSCDs, which is done for all measurements taken.



### 3.7.2 Computation of path-averaged NO₂ mixing ratios

A modified geometrical approach (MGA) for estimating long-path averaged mixing ratios of trace gases (e.g. $NO_2$) from MAX-DOAS measurements at high-altitude sites was proposed in a recent study by Gomez et al. (2014). The method assumes a single-scattering geometry and a scattering point altitude close to that of the instrument. Under these assumptions, the slant paths of the zenith ($\alpha = 90°$) and horizontal ($\alpha = 0°$) measurements are identical up to the scattering point and thus, cancel in the DSCD when using a zenith-sky background spectrum close in time. For measurements performed at higher altitudes, the MGA can be applied without any correction factors, in particular when the instrument is located well above the PBL and aerosol amounts are negligibly low (Schreier et al., 2016). For MAX-DOAS measurements carried out close to the ground level, however, the MGA is limited because of a substantial aerosol load and correction factors are needed (Sinnreich et al., 2013). Nevertheless, Seyler et al. (2017) have recently successfully utilized the MGA for MAX-DOAS measurements of shipping emissions in the German Bight – without the use of correction factors. According to their findings, typical lengths of horizontal light paths in the visible spectral range are in the range of 12.9±4.5 km on average and can reach up to 15 km on days with optimal visibility. It should be noted, however, that the non-consideration of correction factors in polluted environments such as the German Bight will lead to a systematic overestimation of horizontal path lengths, depending on the aerosol load.

In our study, where the rotating tower platform is also located close to the ground level, we overcome this problem by making the following assumptions. Firstly, we assume that the signal for horizontal measurements ($\alpha = 0°$) is dominated by the horizontal part of the light path after the last scattering event. Secondly, a hill named Kahlenberg (484 m a.s.l.) and being located in the Northwest of the Vienna Danube Tower (305°) comes into field of view once every rotation. We assume that the hill limits the horizontal optical path length (hOPL) under clear sky conditions and use the distance between the summit of the hill and the Vienna Danube Tower (6.95 km) as normalization value. The conversion of DSCD $O_4$ at $\alpha = 0°$ is realized by relating this distance with the obtained DSCD $O_4$ value at 305° and applying the resulting relationship to all other DSCD $O_4$ values observed during the same tower platform rotation. We assume that the change of DSCD



NO$_2$ in the vertical ($\alpha = 90°$) can be neglected for (polluted) urban environments over the course
of one tower rotation. The latter assumption has to be made because no sequential zenith-sky
spectra are available. Therefore, path-averaged NO$_2$ mixing ratios are only estimated and presented
for the last tower rotations of the individual days, having the zenith-sky reference spectrum as close
as possible in time.
When taking all these assumptions into consideration, path-averaged mixing ratios of NO$_2$ can be
estimated with the following equation:
$Tower\ DOAS\ X_{NO_2} = \left(\frac{DSCD\ NO_2}{hOPL}\right)/n_a$                          (6)
For the calculation of $n_a$, rotation averages of pressure and temperature as provided by the BOKU
weather station are used (see Sect. 3.5).
**4 Results and discussion**
**4.1 Horizontal gradients of NO$_2$ DSCDs**
As the car DOAS zenith-sky measurements provide in addition to the temporal distribution the
horizontal variation of NO$_2$, the method described in Sect. 3.2.1 is applied to the car DOAS zenith-
sky observations to determine horizontal gradients of NO$_2$.
In Figure 7, typical examples of such horizontal gradients are presented for 27 September, 6
October and 3 November 2015 – three days with different wind conditions, temperature levels and
tropospheric NO$_2$ amounts (see Table 2). In general, an increase in absolute NO$_2$ differences with
increasing distance from the individual starting points is found. While absolute NO$_2$ differences
sharply increase within the first one or two kilometers for most of the journeys, the increase
significantly weakens during the remaining kilometers. During the first kilometer, absolute NO$_2$
differences increase by a factor of 1.5 to 4, depending on the overall NO$_2$ level on the investigated
days. While the absolute NO$_2$ differences rise by a factor of about two within the first two
kilometers on 27 September, an increase by a factor of almost four is found for the same distance
on the more polluted 6 October 2015.





The results imply that the magnitude of absolute $NO_2$ differences is linked to the magnitude of
tropospheric $NO_2$ amounts observed. On the other hand, it is difficult to detect the factors affecting
the shape of the derived curves. Interestingly, we found only small differences in the shape and
magnitude of horizontal $NO_2$ gradients when comparing individual car journeys of single days with
each other. Only for days with significant changes in wind direction (e.g. 27 September 2015) are
the differences in magnitude obvious, when the single laps are compared with each other. While
the curves of 10 April (not shown) and 6 October are similar in shape, the typical sharp increase
within the first two kilometers is not observed for 3 November, although average values of wind
speed, wind direction and mixing-height were similar on those days (see Table 2). It is not clear
why the shape of $NO_2$ as a function of distance observed on 3 November differs from those found
on the other two days. One reason could be variations in photochemistry and/or emissions and/or
dilution of $NO_x$. It is interesting to note that 3 November 2015 was clearly the coldest day with
temperatures below 5°C (see Table 2). As a result, we argue that the characteristic horizontal $NO_2$
scale of the observed $NO_2$ fields in Vienna is on the order of 1 to 2 km.
**4.2 Temporal evolution of tropospheric $NO_2$**
Figure 8 shows typical car DOAS zenith-sky measurements of $NO_2$ performed on 10 April 2015.
The black and red curves represent $DSCD_{meas}$ and $VCD_{tropo}$, respectively. The stratospheric $NO_2$
amounts as simulated by B3dCTM and scaled to GOME-2 observations (see Sect. 3.2.3) are
illustrated by the blue line. Clearly, stratospheric $NO_2$ is relatively low in this case of increased
tropospheric $NO_2$ levels when compared to $VCD_{tropo}$. The relatively small diurnal increase of $NO_2$
in the stratosphere can hardly be seen for the 6-hour period. There are individual peaks in $NO_2$
throughout the morning of 10 April 2015. While the longer lasting $NO_2$ peaks are probably
connected to pollution from traffic, sharp peaks rather indicate some outflow of $NO_2$ from the
refinery and/or other local static emission sources. The magnitude of observed $NO_2$ $VCD_{tropo}$ is in
good agreement with measurements performed around the German cities Mannheim and
Ludwigshafen as well as in the Romanian city Braila (Ibrahim et al., 2010; Dragomir et al., 2015).
As expected, significantly higher values of $NO_2$ $VCD_{tropo}$ were observed by Wang et al. (2012) in
the central urban area of Shanghai, China.





In the following, the small-scale transport of $NO_2$ is evaluated along the Donauufer motorway
(A22) in more detail. The A22 motorway, which is identifiable in Fig. 1 by azure blue and turquois
dots (NW to SE), is one of the busiest roads in Vienna, in particular in the south-eastern area, where
many commuters take the Südosttangente motorway (A23) at the motorway junction
Kaisermühlen. The A23 is another busy road in Austria having about 160000 passenger cars
driving every day on average (www.vcoe.at). As a consequence, $NO_2$ levels are expected to be
significantly increased in this area, in particular during the morning and evening rush hours.
The $NO_2$ variation along the A22 motorway is shown in Fig. 9 for Friday, 10 April and Friday, 3
November 2015 as a function of cumulative distance, where the starting and end points are in the
NW and SE of the A22 motorway. The red, blue, and green curves represent $NO_2$ $VCD_{tropo}$ during
the first, second, and third drive, respectively. In order to not confuse the reader, the first and second
rounds of days with measurements taken only during two rounds are here referred to as round two
and three, starting approximately at 07:00 and 08:30 UT, respectively (see Table 2). While wind
was blowing from SE on both days, averages of wind speed were slightly higher on 3 November.
On 10 April, highest $NO_2$ $VCD_{tropo}$ is observed in the SE rather than in the NW during the first
drive. This seems reasonable as the traffic volume is generally largest in this area, in particular
during the morning rush-hour, which is captured by the first drive of that day. $NO_2$ loads are then
moving to the NW of the A22 motorway, because air masses are transported from SE. A clear shift
of $NO_2$ pollution from SE to NW is observed on 10 April 2015. Highest $NO_2$ $VCD_{tropo}$ during the
first (~2.3 x $10^{16}$ molec cm$^{-2}$), second (<2.5 x $10^{16}$ molec cm$^{-2}$), and third drive (>2.5 x $10^{16}$ molec
cm$^{-2}$) are located around 19.5, 18.5, and 8.5 km away from the starting point in the NW,
respectively. Interestingly, the observed $NO_2$ peak during the last drive is very pronounced. We
attribute this to the $NO_2$ formation via the chemical reaction of NO with ozone towards noon time.
The topography in this area could also be responsible for these high $NO_2$ levels. There are two hills
left (Bisamberg, 358 m a.s.l.) and right (Kahlenberg, 484 m a.s.l.) of the Danube River. As a
consequence, the pollution load could be channeled between the two hills, leading to a localized
increase in $NO_2$ amounts in this area.
The distance of $NO_2$ transport appears larger between the second and third drives when compared
with distances of $NO_2$ transport between the first and second journey. This might be related to the





increase in average wind speed throughout the morning (see Table 2). Overall, the distance of $NO_2$
transport on 10 April 2015 is in good agreement with average wind speed. Due to higher wind
speeds on 3 November 2015, the expected peaks of $NO_2$ in the NW during the third journey cannot
be seen anymore. This might be related to the high averaged wind speeds during the second and
third drives (between 8 and 10 km h$^{-1}$) and thus, a distance of transport exceeding the area of
evaluation. On the other hand, a clear shift of elevated $NO_2$ amounts into the NW is also observed
for the second round on 3 November 2015. It is interesting to note that the horizontal extent of
elevated $NO_2$ amounts during the third round of 10 April and during the second round of 3
November 2015 spans about 8 km in both cases – under similar wind speeds. We argue that this is
a characteristic horizontal extent of a $NO_2$ plume resulting from morning rush-hour traffic in
Vienna under calm southeasterly winds.
The spatial and temporal variation in tropospheric $NO_2$ amounts is also evaluated by analyzing the
tower DOAS off-axis measurements. In order to correct light path lengths in the troposphere, $NO_2$
DSCDs are normalized with $O_4$ DSCDs. When looking at the time series of intensity (see Fig. 2),
$NO_2$, and $O_4$ (Fig. 10), it becomes apparent that these parameters show variations as a function of
azimuth angle. This variation is repeated with each further tower platform rotation. Although some
similarity is found between DSCD $NO_2$ and $O_4$, the highest and lowest amounts of both trace gases
are somehow shifted on the x-axis. Some similarity between DSCD $O_4$ and $NO_2$, which is observed
on all five days (not shown), is attributed to changes in the light path. Interestingly, the
normalization with $O_4$ slightly changes the azimuthal position of the pollution peaks towards the
city center.
The geographical distribution of DSCD $NO_2/O_4$ is shown in Fig. 11 for 10 May 2016, when tower
DOAS off-axis measurements during nine platform rotations were collected. The values plotted on
the map are mean $NO_2/O_4$ values and the radius is the $O_4$ column. On that day, wind was mainly
blowing from easterly to southeasterly directions. As a result, highest $NO_2/O_4$ ratios are observed
towards the city center.
The spatial and temporal variability of DSCD $NO_2/O_4$ as obtained from tower DOAS off-axis
measurements is shown in Fig. 12 for 9 and 10 May 2016. As already identified from the analysis
of the car DOAS zenith-sky measurements, highest tropospheric $NO_2$ over Vienna is found in the




1. early morning – a consequence of both a lower (nocturnal) mixing-height and emissions of $NO_x$

2. from morning rush hour traffic. Highest $NO_2$ amounts on both days are generally observed over

3. the city center of Vienna, which is located to the Southwest of the Vienna Danube Tower. A closer

4. look suggests that DSCD $NO_2/O_4$ is about a factor two larger on 9 May than on the 10 May. While

5. wind was constantly blowing from the SE on both days, the explanation for this is most likely the

6. higher wind speeds on 10 May.

7.

## 4.3 Comparison of $NO_2$ from car DOAS zenith-sky measurements with in situ $NO_2$

9. The spatial and temporal evolution of $NO_2$ on 10 April 2015 in Vienna as observed by car DOAS

10. zenith-sky (dots) and in situ measurements (squares) is shown in Fig. 13. Wind direction and wind

11. speed obtained from local weather stations are indicated by white arrows. The geographical maps

12. illustrate the spatial distribution of tropospheric $NO_2$ during the three performed journeys on that

13. day. As already highlighted in Sect. 4.2, a clear change in the amount of $NO_2$ throughout the

14. morning is observed along the motorway A22. A large proportion of observed $NO_2$ amounts is

15. produced from traffic emissions of $NO_x$ during the morning rush-hour traffic, in particular in the

16. area southeast of the city center. During the time period of about 4.5 hours between starting and

17. end point of the measurements performed on that day, $NO_2$ is transported over a distance between

18. 10 and 15 km. Another hotspot of increased $NO_2$ levels is observed close to an oil refinery in the

19. SE. The outflow of the refinery is in good agreement with wind direction on that day. As already

20. mentioned in Sect. 4.2, such peaks of $NO_2$ amounts as a result of local static emission sources are

21. sharper than those originating from typical rush-hour traffic. There is a clear decrease of

22. tropospheric $NO_2$ throughout the morning (see also Table 2), most likely as a consequence of

23. dilution and/or the reaction of $NO_2$ with the hydroxyl radical (OH), which is the largest $NO_x$ sink

24. during daytime.

25. Overall, averages of tropospheric $NO_2$ observations were highest on 10 April 2015 and 3 November

26. 2015. We attribute this behavior to the comparatively low wind speeds, and consequent low

27. dilution.



As outlined in Sect. 3.5, the correlation of the two data sets (car DOAS zenith-sky versus in situ)
uses data converted into $NO_2$ mixing ratios, which are gridded values onto 0.01° x 0.01° cells. The
correlation is performed for each single day where car DOAS zenith-sky measurements were
carried out. The scatter plots including statistics about slope, intercept and correlation coefficient
are illustrated in Fig. 14. Each of the diamonds represents a grid box average of $X_{NO2}$ from car
DOAS zenith-sky measurements as a function of averaged $X_{NO2}$ concentrations from in situ
monitors. The correlation coefficient on 10 April 2015, for example, is 0.8, suggesting a close
linear relationship of the two independent $NO_2$ measurements on that day (see also Table 2). The
negative offset apparent implies that in situ $X_{NO2}$ is higher than $X_{NO2}$ estimated via Eq. 3. While
this is the case for the grid box averages calculated from measurements taken during the second
and third journeys of that day, $X_{NO2}$ from car DOAS zenith-sky observations seem to be
overestimated during the first journey. $X_{NO2}$ values close to the 1:1 line are also observed on 2
October, the second day, when early morning measurements were performed and when wind was
also blowing from Southeast. The reason for the better agreement in the early morning (e.g. during
the first car journey) could be the lower MH and lower wind speed, resulting in a better vertical
mixing within the shallow boundary layer. The increase in both MH and wind speed throughout
the morning might counteract a vertical mixing of $NO_2$ loads.
Another explanation of the rather underestimated mixing ratio values obtained from car DOAS
zenith-sky measurements observed on the other days is a possible overestimation of tropospheric
AMFs, which are used for the conversion of $NO_2$ DSCDs (see Eq. 1). Wang et al. (2012) have
reported total uncertainties of tropospheric AMFs in the range of 20-30% for SZAs<40°. With
increasing SZA towards sunrise/sunset the uncertainties further increase. We note that most of our
car DOAS zenith-sky measurements were performed for SZAs larger than 40°.
Kramer et al. (2008) performed a comparison between data from a Concurrent MAX-DOAS
(CMAX-DOAS) instrument and in situ instruments in the city of Leicester, England. They
highlighted the fact that the relative positions of the in situ instruments to the streets affect the
comparison. In contrast to their study, car DOAS zenith-sky measurements were performed along
motorways in our study. Therefore, this effect can be partly ruled out for the comparison presented
in our study. Difficulties rather arise from losing some of the $NO_2$ signal at the surface levels
because of the zenith-sky geometry applied for our car DOAS measurements.





Nevertheless, large correlation coefficients (R = 0.72-0.94) are also observed on the other days
with wind coming from the SE (6, 27 October, and 3 November). In contrast, weak correlation
between the two data sets is observed on days when wind was blowing from the NW (27
September, 19 and 23 October). The reason for the weak correlation on those days is not entirely
clear. However, a closer look reveals that the variability of $NO_2$ levels between the performed car
journeys on a single day is only low on days with winds from NW (see Table 2). This might be
related to the fact that high traffic volume but also most of the in situ monitoring station used in
this study are located rather in the SE of the city center than in the NW and thus, the peak of rush-
hour traffic does not show up in the measurements of most of the in situ monitoring stations on
those days.
As $X_{NO2}$ estimated via Eq. 3 represent averages within the PBL and thus, values are rather
underestimated when compared to the values obtained from air quality monitoring stations (see
Table 2), a linear regression analysis is introduced (see Eq. 4). The motivation behind this approach
is related to the findings of Dieudonne et al. (2013). The authors of that study highlighted the fact
that the vertical distribution of $NO_2$ within the PBL over an urban area is not homogenous. They
also suggested considering the effect of wind speed on the vertical gradient. Therefore, we also
include wind speed in the linear regression analysis.
The lap averages of Car DOAS (Surface) $X_{NO2}$ are given in Table 2. Overall, the values are in good
agreement with the lap averages obtained from the air quality monitoring stations. For a better
view, the modeled mixing ratios are plotted against mixing ratios obtained from in situ
measurements in Fig. 15. The gray dotted lines represent the ±25% level, meaning that all the
values estimated via Eq. 4 are within ±25%, with the exception of values lower than 10 ppb. The
reason for these larger differences could be a reduced signal-to-noise of the car DOAS zenith-sky
measurements and consequently larger errors in the $NO_2$ DSCDs. Nevertheless, the high correlation
coefficient of the linear relationship (R = 0.94) is promising, in particular when thinking of
applying this method to $NO_2$ $VCD_{tropo}$ obtained from long-term MAX-DOAS measurements,
which provide better statistics.
**4.4 Path-averaged NO₂ mixing ratios**



Although the $NO_2/O_4$ ratio gives an overall impression of spatiotemporal changes of $NO_2$ amounts
over Vienna, an absolute quantification of $NO_2$ amounts (e.g. the conversion into mixing ratios) is
not possible with this approach. Therefore, another method is used for the estimation of path-
averaged $NO_2$ mixing ratios at 160 m altitude of the rotating tower platform (see Sect. 3.7.2).
Estimated horizontal optical path lengths as a function of the azimuthal viewing direction obtained
from measurements taken on 29 April (blue) and 9 May (red) 2016 are shown in Fig. 16. Both
curves represent the last round measurements recorded during those days, when the reference
zenith-sky measurement was taken shortly afterwards. Overall, higher hOPLs are observed on 9
May, which was a day with wind speeds reaching up to 15 km h$^{-1}$. The exceptionally low wind
speeds observed on 29 April ($<$ 5 km h$^{-1}$) explain the lower values of hOPL on that day. Low values
of hOPL are generally linked to low visibility, which is the result of an increased aerosol
accumulation over emission hot spots on that otherwise cloudless day. As aerosols largely affect
hOPL under such conditions (Sinreich et al., 2013), it is reasonable that lowest values (5-6 km) are
preferably found in off-axis directions between Eastern and Southern directions, which include
areas with high traffic roads and industry. In contrast, highest hOPLs are observed in the North of
the Vienna Danube Tower on both days (10-11 km). This is reasonable because those regions are
known as rather rural areas without significant emission sources. The highest hOPLs estimated in
our study are slightly lower than the mean value (12.9 km) reported in Seyler et al. (2017), but still
within the standard deviation.
Although our assumption made on the limitation of the horizontal light path length towards the hill
might be critical, we argue that the distance of 6.95 km between the Vienna Danube Tower and the
summit of that hill is still lower than 12.9±4.5 km and thus seems to be optimal for this
normalization approach.
Estimated path-averaged $NO_2$ mixing ratios are shown for 29 April (blue) and 9 May (red) 2016 in
Fig. 17. Again, only the last rotations of those days are presented in the graph. As expected from
the observed wind conditions and estimated hOPLs, path-averaged $X_{NO2}$ is higher on 29 April.
Over rural areas, which are located in the North of the Vienna Danube Tower, values are lowest
(2.5 to 4 ppb) on both days. In contrast, highest values (up to 9 ppb) are again observed towards



SE. We note that path-averaged mixing ratios are only shown for two tower rotations, which took
place shortly before noon – at a time when the peak in $NO_2$ amounts over the city is past.
For a better illustration, $X_{NO2}$ as a function of hOPL obtained from the last rotation of tower DOAS
off-axis measurements and $X_{NO2}$ values calculated from simultaneous in situ measurements are
plotted on a geographical map in Fig. 18 for 29 April (left) and 9 May 2016 (right). We note that
the $NO_2$ mixing ratios estimated from tower DOAS off-axis measurements are averages over
several kilometers at 160 m above ground, whereas $NO_2$ mixing ratios from in situ stations rather
represent point measurements at the surface level. The comparison therefore implies that the
variability of $NO_2$ as observed at 160 m above ground is much less pronounced than that between
the individual ground stations. Moreover, horizontal gradients in 160 m above ground are small.
As already outlined above, highest $NO_2$ amounts obtained from both measurements are generally
found over the city center and over high traffic roads in the Southeast of the city center on 29 April,
a day with very low wind speeds ($< 5km\ h^{-1}$). The picture looks different for 9 May, when wind
was blowing from Southeast and wind speeds reached values of up to 15 km $h^{-1}$. Highest $NO_2$
amounts from tower DOAS off-axis observations are found in parallel to the wind direction in this
case. On both days, $NO_2$ mixing ratios are about a factor four larger at the surface level when
compared with path-averaged values at 160 m above, which is in good agreement with the 25%
reported in Dieudonne et al. (2013), who compared surface concentrations with in situ
concentrations at 300 m above ground in Paris.
**5 Summary and outlook**
In this case study, unique ground-based remote sensing measurements have been coupled with
surface in situ measurements to investigate the $NO_2$ distributions in the planetary boundary layer
in the Viennese metropolitan area.
A DOAS instrument was used for the determination of the spatial and temporal $NO_2$ distributions
in and around the urban area of Vienna. The instrument was applied in two different measurement
setups: Car DOAS zenith-sky and tower DOAS off-axis. The former DOAS-type approach, which
is already well established and documented in the literature, was used for a total of twenty identical



car journeys, which were carried out on nine days in April, September, October, and November
2015 during the morning hours. The latter configuration is innovative in the sense that horizontal
measurements for more than 100 azimuthal angles are possible within a 360° rotation and within
less than half an hour. The latter setup was used for collecting more than thirty rotations of spectral
measurements on five days in April and May 2016.
A DOAS fitting procedure, based on recommendations made for the CINDI-2 campaign
(www.tropomi.eu/data-products/cindi-2), is applied to the collected spectral measurements to
retrieve $NO_2$ DSCDs. Overall, good fit quality is found for both DOAS-type measurements, in
particular when $NO_2$ amounts were high.
As the car DOAS zenith-sky measurements include a contribution from both the background and
stratospheric $NO_2$, a correction scheme based on measurements and chemical transport model
simulations is applied. The subsequent conversion of $NO_2$ DSCDs into $NO_2$ $VCD_{tropo}$ is performed
by applying stratospheric and tropospheric AMFs as derived from radiative transfer calculations.
In order to correct light path lengths in the troposphere, $NO_2$ DSCDs obtained from tower DOAS
off-axis observations are normalized with $O_4$ DSCDs in a first step. In a second step, the assumption
that the Kahlenberg (484 m a.s.l) limits the horizontal optical light path length at an azimuth angle
of 305° is made. The distance between the Vienna Danube Tower and the summit of Kahlenberg
(6.95 km) is then used for the normalization of $O_4$ DSCDs to obtain horizontal optical path lengths
(hOPLs).
By analyzing $NO_2$ DSCDs at high temporal resolution along the individual car journeys,
characteristic horizontal $NO_2$ changes as a function of distance could be derived. While the absolute
differences between the first and consecutive measurements increases sharply over the first two
kilometers (by a factor of 1.5 to 4), the observed increase clearly weakens during the remaining
kilometers. From this observation we conclude, that 1-2 km is a characteristic scale of the $NO_2$
fields observed in Vienna during the morning hours.
The analysis of $NO_2$ $VCD_{tropo}$ from car DOAS zenith-sky and DSCD $NO_2/O_4$ from tower DOAS
off-axis measurements opened up interesting insights into the spatial and temporal variations of
$NO_2$. The results imply that wind speed and wind direction impact strongly on the $NO_2$ distributions



in Vienna. By using data on wind speed and wind direction from several stations within the
metropolitan area of Vienna, short-scale $NO_2$ transport events could be identified.
The comparison of $VCD_{tropo}$ from car DOAS zenith-sky measurements with in situ $NO_2$
concentrations, which is based on the conversion of both quantities into mixing ratios of $NO_2$,
revealed good linear correlation for days when the wind was blowing from the Southeast (R = 0.72-
0.94). In contrast, weak correlation was found for days when the wind was blowing from the
Northwest (R < 0.33), which might be related to the relative location of air masses affected by
dense traffic to the selected in situ monitoring stations.
Depending on wind conditions, lap averages of near-surface $NO_2$ mixing ratios ($XNO_2$) estimated
from car DOAS zenith-sky measurements applying a linear regression analysis are in the range of
3.8 to 26.2 ppb and in good agreement with $XNO_2$ obtained from in situ measurements.
Taking into account all the assumptions that have been made for the conversion of DSCDs into
$VCD_{tropo}$ and also for the subsequent translation of $VCD_{tropo}$ into $X_{NO2}$, the method to derive near-
surface mixing ratios seems to work well – at least for the lap averages considered in this study.
The estimation of hOPL and $X_{NO2}$ from the tower DOAS off-axis measurements revealed
interesting insights into an upper layer of the PBL, although only few measurements are presented
due to the lack of sequential zenith-sky measurements that could be taken as reference. Overall,
$NO_2$ mixing ratios are about a factor four larger at the surface level when compared with path-
averaged values at 160 m above. The path-averaged mixing ratios are about 35% smaller at 160 m
above ground, when qualitatively compared to $X_{NO2}$ from car DOAS zenith-sky measurements
performed on days with similar wind conditions.
Although the $NO_2$ hourly European maximum dose rate was not exceeded when measurements
were taken, $NO_2$ amounts in the urban environment of Vienna are substantial, in particular during
morning hours and when wind speeds are low.
We note that the idea of performing tower DOAS off-axis measurements was born when car DOAS
zenith-sky measurements were already taken. Due to other priorities and limited manpower at the
time when tower DOAS off-axis measurements were recorded, car DOAS zenith-sky



measurements could not be carried out simultaneously. For future campaigns in Vienna, however,
simultaneous measurements of the two DOAS configurations should be taken into consideration.
Future efforts will be made to test the linear regression analysis on measurements from three static
MAX-DOAS instruments, which are located in Vienna as part of the VINDOBONA (VIenna
horizontal aNd vertical Distribution OBservations Of Nitrogen dioxide and Aerosols) project
(www.doas-vindobona.at). Once the method is mature and optimized, it could also be applied to
satellite measurements of $VCD_{tropo}$. This would help to obtain near-surface mixing ratios of $NO_2$
from the integrated column amounts on a global scale.
Additional car DOAS zenith-sky and tower DOAS off-axis measurements that complement the
operational performance of the two MAX-DOAS instruments are also foreseen in the future.
Taking these measurements and also data from the relatively large number of air quality monitoring
stations into consideration, Vienna can be seen as an optimal urban location for future satellite
validation campaigns.

## Acknowledgements

This study was funded by the University of Bremen and the Austrian Science Fund (FWF): I 2296-
N29. We like to thank "Amt der Niederösterreichischen Landesregierung" and "Amt der Wiener
Landesregierung" for making the air quality data freely available. We wish to acknowledge the
provision of meteorological data by the Austrian official weather service (ZAMG). Christoph
Lotteraner and Martin Piringer (ZAMG) are acknowledged for calculating time-series of mixing-
height at Wien/Hohe Warte. We thank Andreas Hilboll (MARUM-Bremen) for the provision of
simulated stratospheric $NO_2$ amounts. Last but not least, we want to thank Mario Meyer and the
staff from the Vienna Danube Tower for giving us the opportunity to perform experimental
measurements from the rotating Café and for providing technical assistance.

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

| Fit parameter | | Selection/Source |
|---|---|---|
| Spectral range | | 425-490 nm |
| Polynomial degree | | 5 (car zenith-sky), 7 (tower off-axis) |
| Wavelength calibration | | Solar atlas (Kurucz et al., 1984) |
| Reference | | Zenith-sky spectrum (close to noontime) |
| **Cross section** | **Temperature** | **Data source** |
| $O_3$ | 223 K | Serdyuchenko et al. (2014) with $I_0$ correction |
| $NO_2$ | 298 K | Vandaele et al. (1996) with $I_0$ correction |
| $O_4$ | 293 K | Thalman and Volkamer (2013) |
| $H_2O$ | - | Rothmann et al. (2010) |
| Ring | - | QDOAS (Danckert et al., 2015) |





**Table 2.** Summary of statistics of the individual car journeys including lap averages of wind speed,
wind direction, temperature, pressure, number density of air, mixing-height, in-situ $NO_2$ from
selected air quality monitoring stations, and $NO_2$ $VCD_{tropo}$ from car DOAS measurements.
Converted averaged $NO_2$ mixing ratios for both measurements are also given. The correlation
coefficients (R) obtained from the linear relationship between car DOAS and in situ $NO_2$ are also
shown (further details are given in the text).

| | 10.04.2015[a] | | | 27.09.2015[b] | | 28.09.2015[b] | |
|---|---|---|---|---|---|---|---|
| Car journey (UT) | 05:27-06:59 | 07:06-08:35 | 08:40-10:04 | 07:11-08:42 | 08:42-10:17 | 06:36-08:20 | 08:21-10:05 |
| Wind speed [km h$^{-1}$][d] | 3.9±2.4 | 5.4±2.8 | 6.7±2.4 | 14.4±4.9 | 15.3±5.4 | 16.1±5 | 19.8±6.4 |
| Wind direction [deg][d] | 135.5±29.6 | 126.2±29.3 | 114.1±24.6 | 337.2±7.1 | 240.3±81.5 | 187.1±114.2 | 91.7±99.4 |
| Temperature [°C][e] | 7.4±1 | 10.4±0.8 | 13±0.7 | 12.6±0.3 | 13.7±0.4 | 12.4±0.6 | 14.2±0.5 |
| Pressure [hPa][e] | 994.6±0.1 | 994.7±0 | 994.5±0 | 996.1±0.3 | 996.5±0.1 | 1000.6±0.2 | 1000.7±0.1 |
| Number density of air [molec cm$^{-3}$][f] | 2.568e19 | 2.541e19 | 2.517e19 | 2.525e19 | 2.516e19 | 2.538e19 | 2.522e19 |
| Mixing-height [m][g] | 148.2±28.6 | 311.7±46.4 | 445.7±30.1 | 1103.7±50.2 | 1045.4±30 | 541.3±105.9 | 1161.1±209.7 |
| In situ $NO_2$ [µg m$^{-3}$][h] | 63.3±22.9 | 43.7±23 | 35.5±20.3 | 9.3±5.2 | 8.4±4.5 | 20.4±11.7 | 15.4±10.2 |
| In situ $X_{NO2}$ [ppb][i] | 31.6±11.4 | 22.0±11.6 | 18.1±10.3 | 4.7±2.7 | 4.3±2.3 | 10.3±5.9 | 7.8±5.2 |
| Car DOAS $NO_2$ [$10^{16}$ molec cm$^{-2}$][j] | 1.34±0.49 | 1.02±0.49 | 0.91±0.74 | 0.23±0.09 | 0.15±0.09 | 0.42±0.18 | 0.33±0.28 |
| Car DOAS (BL) $X_{NO2}$ [ppb][k] | 35.2±12.9 | 12.9±6.2 | 8.1±6.6 | 0.8±0.3 | 0.6±0.4 | 3±1.3 | 1.1±0.9 |
| Car DOAS (Surface) $X_{NO2}$ [ppb][l] | 26.2 | 21.5 | 18 | 7.3 | 6.5 | 11 | 3.8 |
| Correlation coefficient[m] | | 0.80 | | 0.37 | | 0.65 | |

**Table 2.** continued.

| | 02.10.2015[b] | | | 06.10.2015[b] | | 19.10.2015[c] | |
|---|---|---|---|---|---|---|---|
| Car journey (UT) | 05:22-06:58 | 07:01-08:29 | 08:29-09:55 | 06:57-08:23 | 08:24-09:57 | 06:57-08:30 | 08:32-09:56 |
| Wind speed [km h$^{-1}$][d] | 4.7±2.1 | 10.9±2.9 | 16.9±4.5 | 8.1±3.3 | 10±3.1 | 8.1±2.9 | 11±2.6 |
| Wind direction [deg][d] | 125.3±40.3 | 134.8±6.9 | 139.8±5.8 | 120.4±7.7 | 122.8±11.6 | 293.3±23.4 | 312.8±7.1 |
| Temperature [°C][e] | 7.7±1.3 | 11.1±1.2 | 14.8±0.9 | 12.9±1 | 14.7±0.2 | 7.7±0.2 | 8.1±0.1 |
| Pressure [hPa][e] | 997.6±0.1 | 997.5±0 | 997±0.2 | 982±0 | 981.9±0.1 | 987±0.1 | 986.8±0 |
| Number density of air [molec cm$^{-3}$][f] | 2.573e19 | 2.542e19 | 2.508e19 | 2.471e19 | 2.486e19 | 2.545e19 | 2.541e19 |
| Mixing-height [m][g] | 206.8±51.6 | 350.4±44.5 | 666.9±136.2 | 381.6±24.8 | 480.4±34.4 | 412.4±23.2 | 374.5±19.1 |
| In situ $NO_2$ [µg m$^{-3}$][h] | 44.1±17.3 | 27.2±10 | 19.9±12.6 | 27.3±9 | 25.4±10.2 | 30.8±10.9 | 30.4±10 |
| In situ $X_{NO2}$ [ppb][i] | 22±8.6 | 13.7±5 | 10.1±6.4 | 14.1±4.6 | 13.2±5.3 | 15.5±5.5 | 15.3±5.1 |
| Car DOAS $NO_2$ [$10^{16}$ molec cm$^{-2}$][j] | 0.74±0.48 | 0.47±0.35 | 0.24±0.2 | 0.65±0.41 | 0.55±0.37 | 0.53±0.35 | 0.5±0.21 |
| Car DOAS (BL) $X_{NO2}$ [ppb][k] | 13.9±9 | 5.3±3.9 | 1.5±1.2 | 6.8±4.2 | 4.5±3.1 | 5.1±3.4 | 5.2±2.2 |
| Car DOAS (Surface) $X_{NO2}$ [ppb][l] | 23.4 | 16 | 7.1 | 13.7 | 12.2 | 17.6 | 15.7 |
| Correlation coefficient[m] | | 0.73 | | 0.78 | | 0.23 | |



**Table 2.** continued.

|  | 23.10.2015[c] | | 27.10.2015[c] | | 03.11.2015[c] | |
|---|---|---|---|---|---|---|
| Car journey (UT) | 06:58-08:46 | 08:47-10:14 | 06:58-08:37 | 08:37-10:02 | 06:44-08:15 | 08:15-09:43 |
| Wind speed [km h$^{-1}$][d] | 13.8±4 | 14±4.2 | 16±5 | 19±5.2 | 8.2±3.2 | 9.9±3.7 |
| Wind direction [deg][d] | 282.6±8 | 294.5±9.5 | 134±7.2 | 137.1±6.7 | 152.1±31 | 157.2±20.9 |
| Temperature [°C][e] | 10±0.3 | 11.1±0.4 | 9±0.2 | 10.6±1 | 3±0.4 | 4.2±0.4 |
| Pressure [hPa][e] | 991.3±0.4 | 992±0.1 | 991.6±0.1 | 991.7±0.1 | 995.7±0.1 | 995.4±0.1 |
| Number density of air [molec cm$^{-3}$][f] | 2.536e+19 | 2.528e+19 | 2.545e+19 | 2.531e+19 | 2.611e+19 | 2.599e+19 |
| Mixing-height [m][g] | 357.5±24.1 | 482.3±50 | 460±14.7 | 631.2±79.7 | 417.4±8.1 | 471.2±25.7 |
| In situ NO$_2$ [µg m$^{-3}$][h] | 26.3±8.2 | 25.4±7.9 | 22.8±10.3 | 18.8±8.6 | 52.7±19.6 | 36.6±18.2 |
| In situ $X_{NO2}$ [ppb][i] | 13.3±4.2 | 12.8±4 | 11.5±5.2 | 9.5±4.3 | 25.9±9.6 | 18.1±9 |
| Car DOAS NO$_2$ [10$^{16}$ molec cm$^{-2}$][j] | 1.41±0.5 | 1±0.51 | 0.27±0.15 | 0.23±0.14 | 0.88±0.51 | 0.72±0.47 |
| Car DOAS (BL) $X_{NO2}$ [ppb][k] | 15.5±5.5 | 8.2±4.1 | 2.3±1.3 | 1.5±0.9 | 8±4.6 | 5.9±3.8 |
| Car DOAS (Surface) $X_{NO2}$ [ppb][l] | 17.1 | 14.3 | 11.5 | 7.4 | 23.3 | 20.3 |
| Correlation coefficient[m] | 0.07 | | 0.72 | | 0.94 | |

[a] Reference measurement taken on 10 April 2015 at 10:49 UT (48° 17' 52.08'' N, 16° 33' 44.64'' E).
[b] Reference measurement taken on 27 September 2015 at 10:17 UT (48° 21' 52.75'' N, 16° 31' 20.24'' E).
[c] Reference measurement taken on 23 October 2015 at 10:14 UT (48° 21' 53.85'' N, 16° 31' 22.48'' E).
[d] Measurements from 9 stations are provided by ZAMG. Values represent lap averages and standard deviations.
[e] Measurements provided by the BOKU weather station. Values represent lap averages and standard deviations.
[f] Calculations are based on the relationship between pressure and temperature measurements. Values represent lap averages.
[g] Measurements provided by ZAMG. Values represent lap averages and standard deviations.
[h] Measurements from 15 stations provided by UBA. Values represent lap averages and standard deviations.
[i] Conversion of mass concentrations into mixing ratios is based on Eq. 5.
[j] Conversion of DSCD$_{meas}$ into VCD$_{tropo}$ is based on Eq. 1.
[k] Conversion of VCD$_{tropo}$ into boundary layer mixing ratios is based on Eq. 3.
[l] Conversion of VCD$_{tropo}$ into surface mixing ratios is based on Eq. 4.
[m] Values represent correlation coefficients between in situ NO$_2$ [ppb] and Car DOAS (BL) NO$_2$ [ppb].



1 **Table 3.** Overview on selected air quality monitoring stations, operated by the Environment

2 Agency Austria.

| | Lower Austria | | | | | | |
|---|---|---|---|---|---|---|---|
| | Klosterneuburg (Wiener Straße) | Klosterneuburg (Wiesentgasse) | Mannswörth (Danubiastraße) | Schwechat | Vösendorf | Wolkersdorf | Gänserndorf |
| Latitudes | 48° 18' 25'' N | 48° 18' 10'' N | 48° 08' 42'' N | 48° 08' 45'' N | 48° 07' 32'' N | 48° 23' 32'' N | 48° 20' 05'' N |
| Longitudes | 16° 19' 35'' E | 16° 19' 17'' E | 16° 30' 40'' E | 16° 28' 37'' E | 16° 19' 60'' E | 16° 31' 20'' E | 16° 43' 50'' E |

| | Vienna | | | | | | | |
|---|---|---|---|---|---|---|---|---|
| | A23 (Wehlistraße) | Belgradplatz | Floridsdorf | Kaiserebersdorf | Laaer Berg | Liesing | Lobau | Stadlau |
| Latitudes | 48° 11' 05'' N | 48° 10' 30'' N | 48° 15' 42'' N | 48° 09' 26'' N | 48° 09' 41'' N | 48° 08' 17'' N | 48° 09' 45'' N | 48° 13' 37'' N |
| Longitudes | 16° 24' 28'' E | 16° 21' 45'' E | 16° 23' 53'' E | 16° 28' 38'' E | 16° 23' 39'' E | 16° 17' 48'' E | 16° 31' 36'' E | 16° 27' 35'' E |



1  **Table 4.** Overview on selected meteorological stations, operated by the Austrian official weather

2  service.

| | | Lower Austria | | |
|---|---|---|---|---|
| | Brunn am Gebirge | Gänserndorf (Stadt) | Gross-Enzersdorf | Wolkersdorf | |
| Latitudes | 48° 06' 25'' N | 48° 20' 16'' N | 48° 11' 59'' N | 48° 22' 49'' N | |
| Longitudes | 16° 16' 12'' E | 16° 42' 49'' E | 16° 33' 33'' E | 16° 30' 27'' E | |
| | | Vienna | | | |
| | Donaufeld | Hohe Warte | Innere Stadt | Stammersdorf | Unterlaa |
| Latitudes | 48° 15' 27'' N | 48° 14' 55'' N | 48° 11' 54'' N | 48° 18' 21'' N | 48° 07' 30'' N |
| Longitudes | 16° 26' 00'' E | 16° 21' 23'' E | 16° 22' 01'' E | 16° 24' 20'' E | 16° 25' 10'' E |



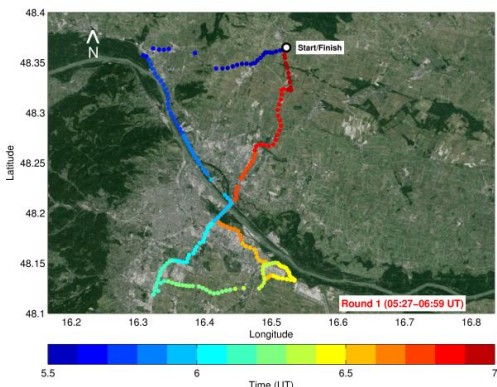

2   **Figure 1.** An example of a single car journey as performed on 10 April 2015 between 05:27 and

3   06:59 UT.





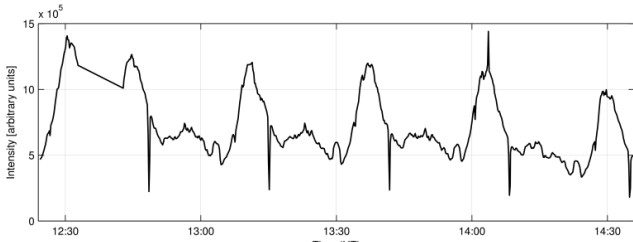

**Figure 2.** An example of a time series of the intensity of the spectrum as measured with the
DOAS instrument from the rotating tower platform on 22 April 2016 between 12:25 and 14:35
UT. The sharp dips indicate a decrease in intensity due to pointing towards a skyscraper, which
blocks the view of the instruments.





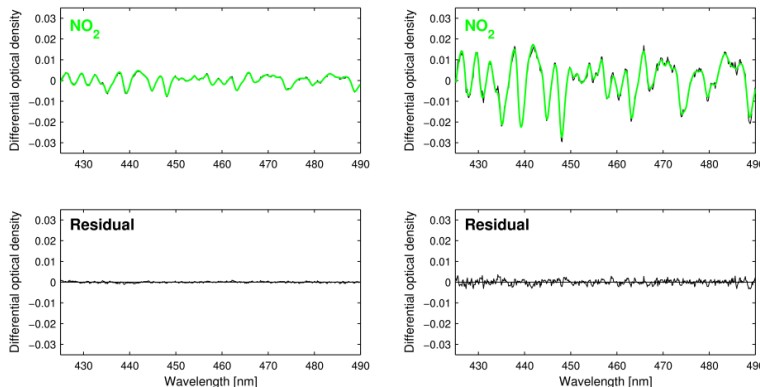

**Figure 3.** Exemplary fit results from the DOAS analysis in the 425-490 nm fitting window for a
car DOAS spectrum (left panels), as measured on 10 April 2015 (SZA = 47.68°, DSCD = 4.03 x
$10^{16}$ molec cm$^{-2}$) and for a tower DOAS spectrum (right panels), as measured on 29 April 2016
(SZA = 66.99°, DSCD = 1.46 x $10^{17}$ molec cm$^{-2}$).





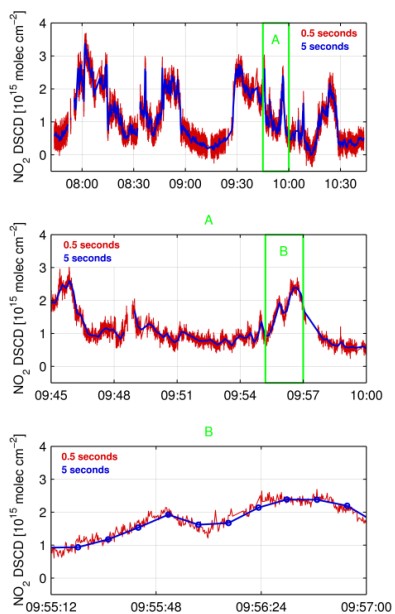

**Figure 4.** Temporal resolution of $NO_2$ DSCDs based on the car DOAS zenith-sky measurements
performed on 3 November 2015. The red and blue lines show data at a resolution of 0.5 and 5
seconds, respectively. The upper panel shows the $NO_2$ DSCDs for the whole period of
observations of that day, whereas the middle and lower panels represent shorter time sections for
clarity.



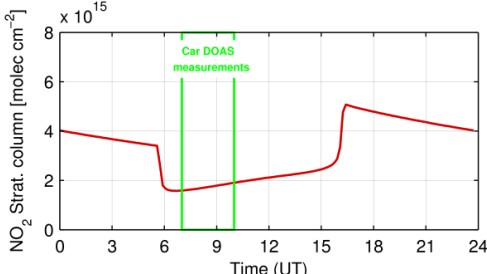

2   **Figure 5.** Stratospheric NO$_2$ above Vienna on 19 October 2014 (red line) as obtained from the

3   Bremen 3d chemistry transport model (B3dCTM). The green rectangle indicates the time period

4   of car DOAS measurements performed on that day.





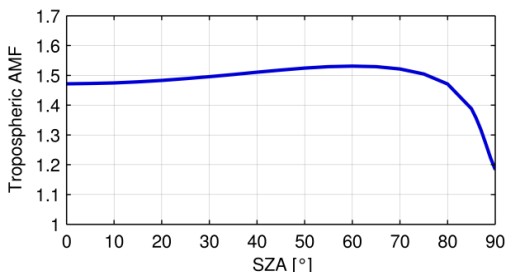 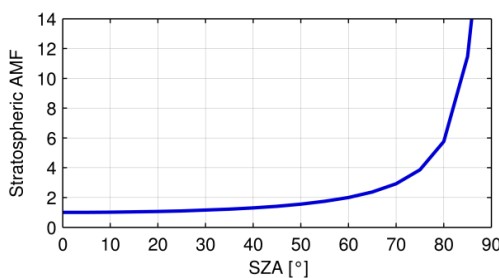

2 **Figure 6.** Computed AMFs as a function of solar zenith angle for the troposhere (left) and

3 stratosphere (right).





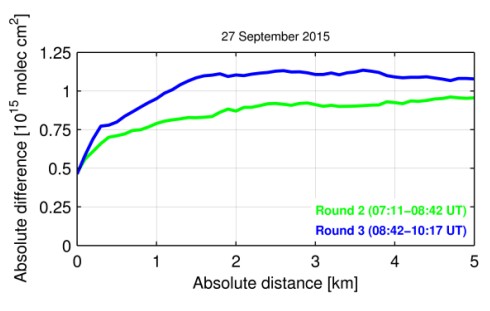   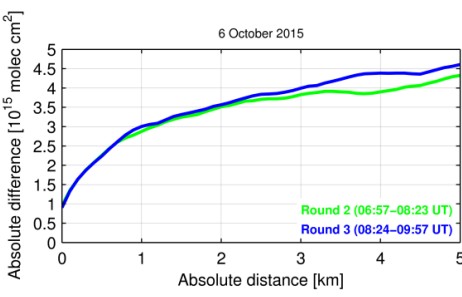

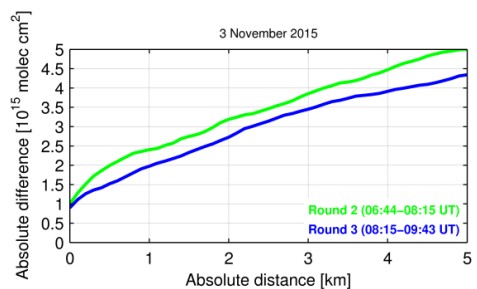

3   **Figure 7.** Mean absolute difference in NO$_2$ DSCDs as a function of the absolute distance (see

4   Sect. 3.2.1) for car DOAS zenith-sky measurements performed on three selected days with

5   different wind conditions and NO$_2$ levels.



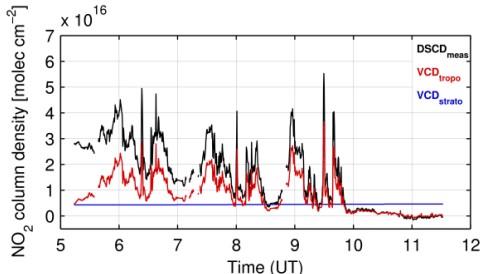

**Figure 8.** Time series of NO$_2$ DSCD$_{meas}$ (black), VCD$_{tropo}$ (red), and VCD$_{strato}$ (blue) obtained

from car DOAS zenith-sky spectra recorded on 10 April 2015.





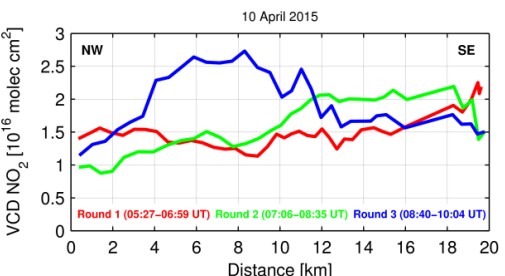 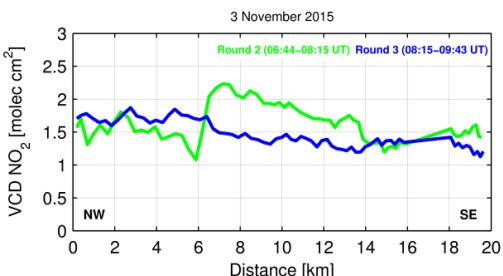

2 **Figure 9.** Temporal evolution of tropospheric $NO_2$ for the car DOAS zenith-sky measurements as

3 performed on 10 April and 3 November 2015. The red, green, and blue curves represent $NO_2$

4 $VCD_{tropo}$ as obtained along the A22 during the first, second, and third journey, respectively.



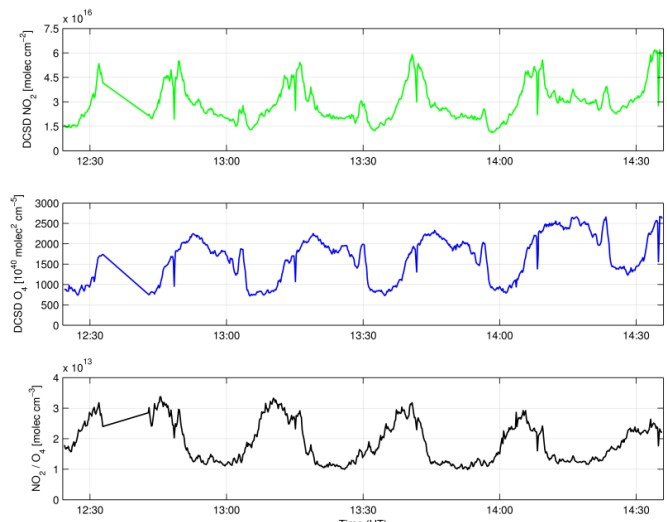

2    **Figure 10.** Time series of NO$_2$ (upper) and O$_4$ (middle) DSCDs as obtained from the tower

3    DOAS off-axis measurements performed on 22 April 2016. The ratio of NO$_2$/O$_4$ is shown in the

4    lowest panel.





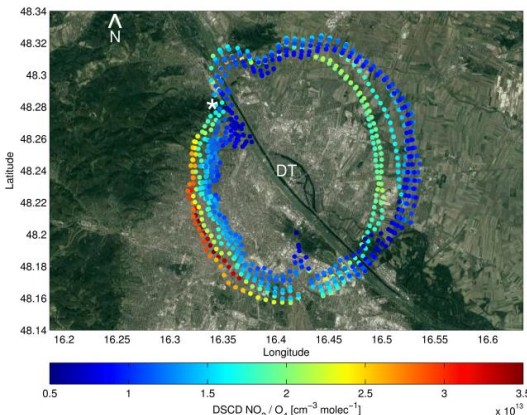

**Figure 11.** Spatial and temporal variability of the $NO_2/O_4$ ratio (here, the radius is determined by

DSCD $O_4$ values) on 10 May 2016 between 05:57 and 09:56 UT observed by tower DOAS off-

axis measurements. The position of the Vienna Danube Tower (DT) is highlighted in the center

of the geographical map. The white asterisk represents the summit of Kahlenberg (484 m a.s.l),

which is used for the estimation of horizontal optical path lengths.





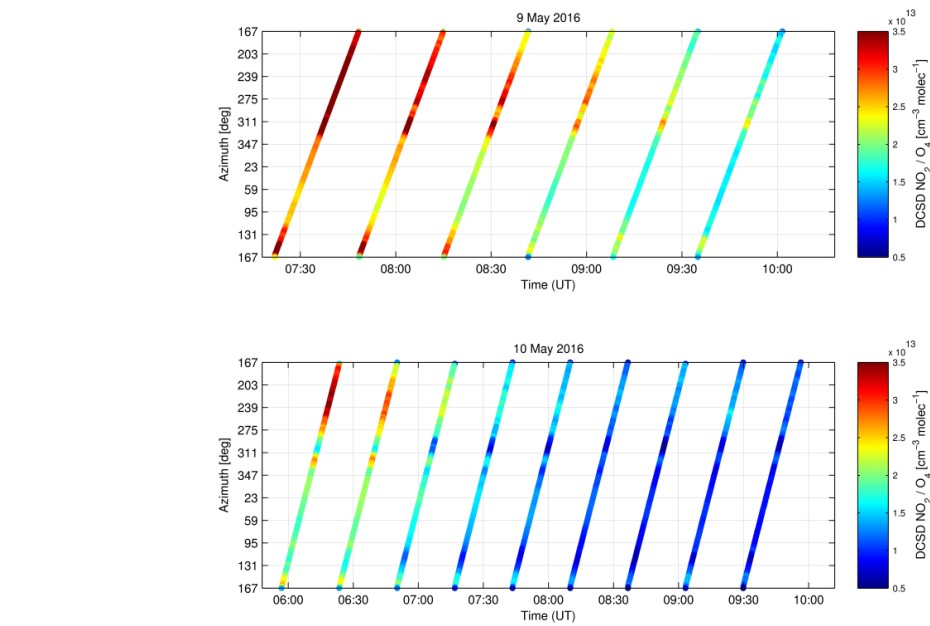

3 **Figure 12.** Spatial and temporal variability of DSCD NO$_2$/O$_4$ obtained from tower DOAS off-

4 axis measurements performed on 9 and 10 May 2016.



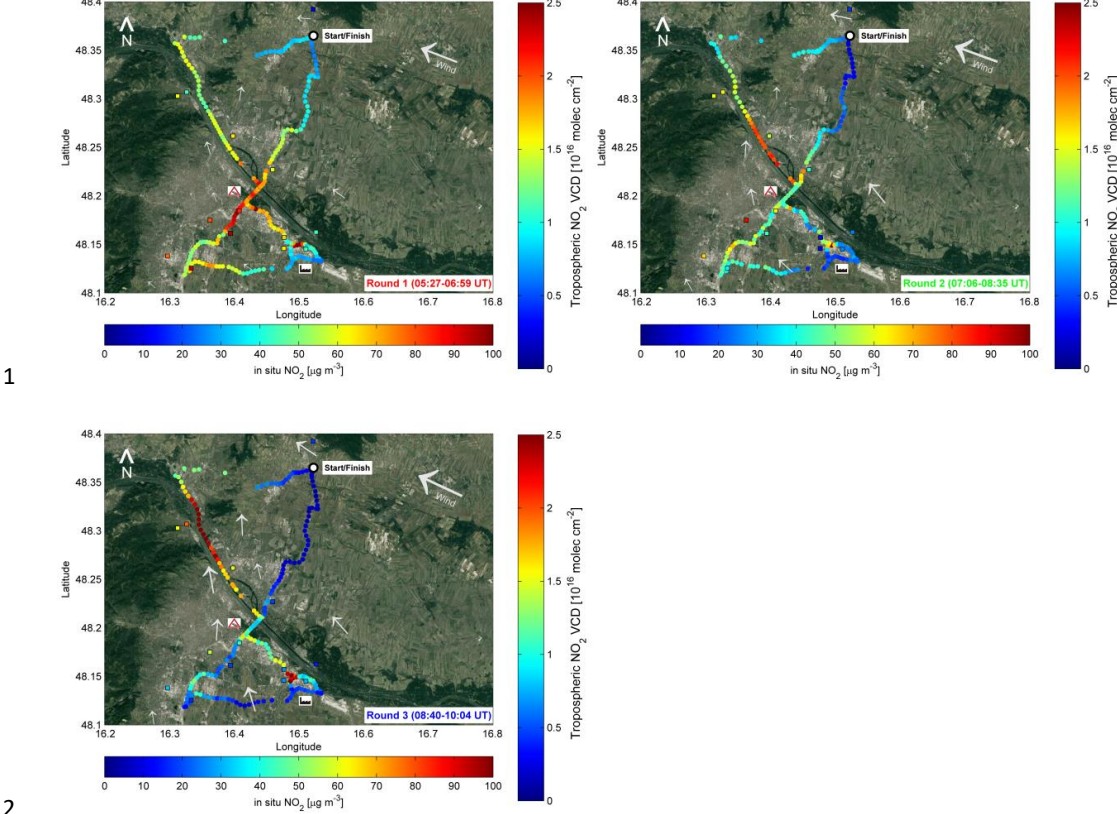

**Figure 13.** Spatial and temporal evolution of $NO_2$ on 10 April 2015 in Vienna as measured by the car DOAS zenith-sky (dots) and in situ surface measurements (squares). Wind direction and wind speed obtained from local weather stations are indicated by white arrows. The size of the arrows is weighted by the corresponding averaged wind speed (2 m above ground) obtained from the individual weather stations. Averaged wind speeds over the course of the car DOAS zenith-sky measurements taken on this day ranged between 2.28 and 12.81 km $h^{-1}$.



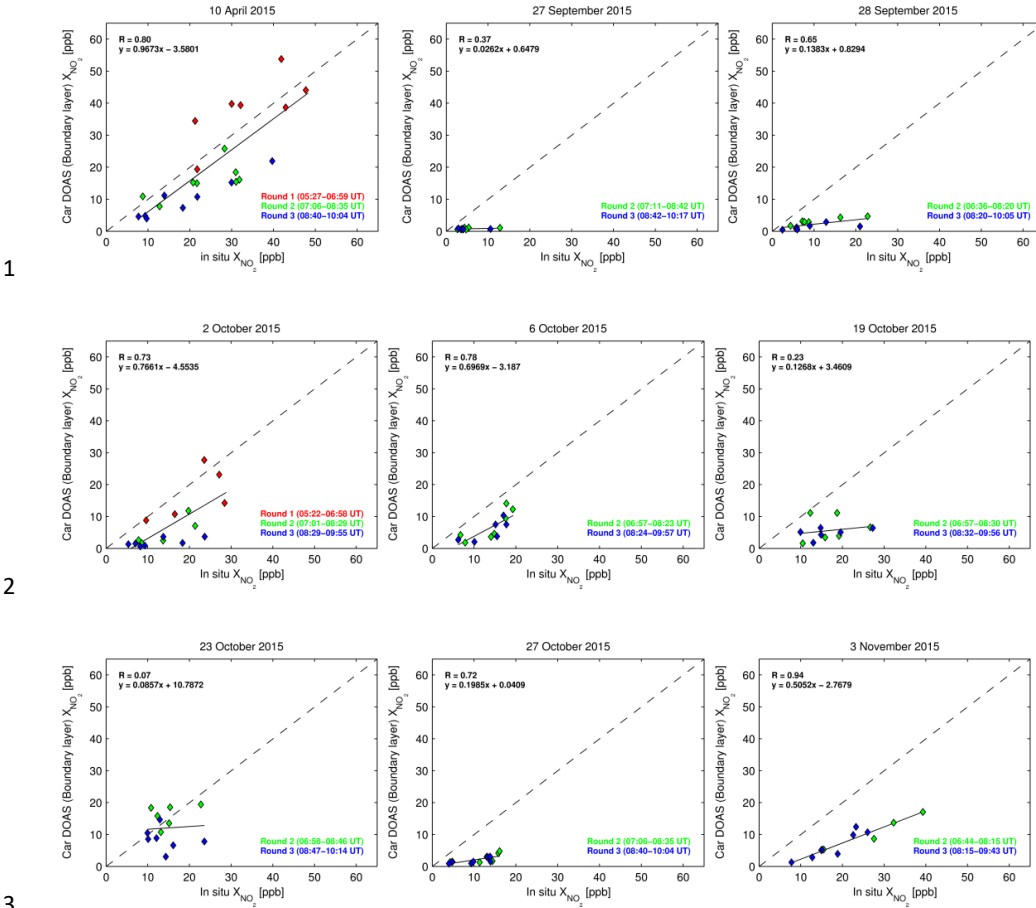

4 **Figure 14.** Comparison of boundary layer NO$_2$ mixing ratios estimated from car DOAS zenith-

5 sky measurements with NO$_2$ mixing ratios obtained from in situ measurements on the nine days

6 when measurements were performed. The dotted line represents the 1:1 relationship.





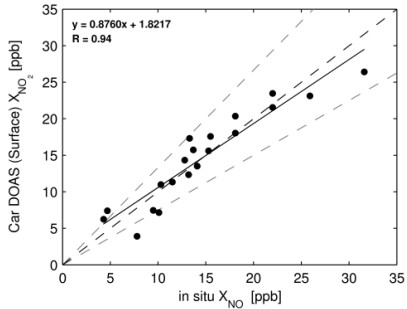

**Figure 15.** Comparison of lap averaged near-surface $NO_2$ mixing ratios estimated from car
DOAS zenith-sky measurements with $NO_2$ mixing ratios obtained from in situ measurements.
Lap averages of all twenty performed car rides are included in the calculation. The black and grey
dotted lines represent the 1:1 relationship and ±25%, respectively.



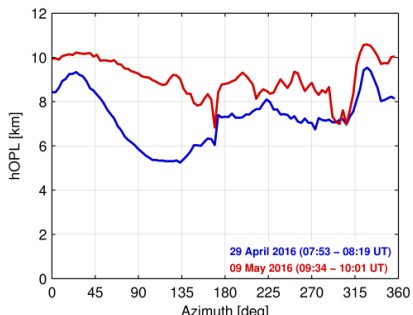

2    **Figure 16.** Estimated horizontal optical path length obtained from tower DOAS off-axis

3    measurements recorded during two tower platform rotations on 29 April and 9 May 2016.





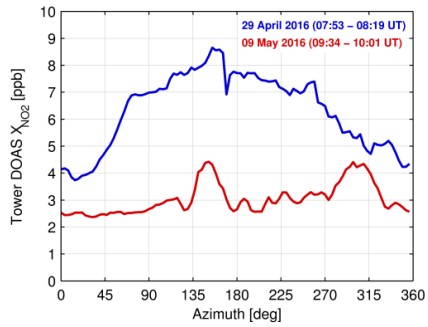

2 **Figure 17.** Estimated path-averaged NO$_2$ mixing ratios obtained from tower DOAS off-axis

3 measurements recorded during two tower platform rotations on 29 April and 9 May 2016.





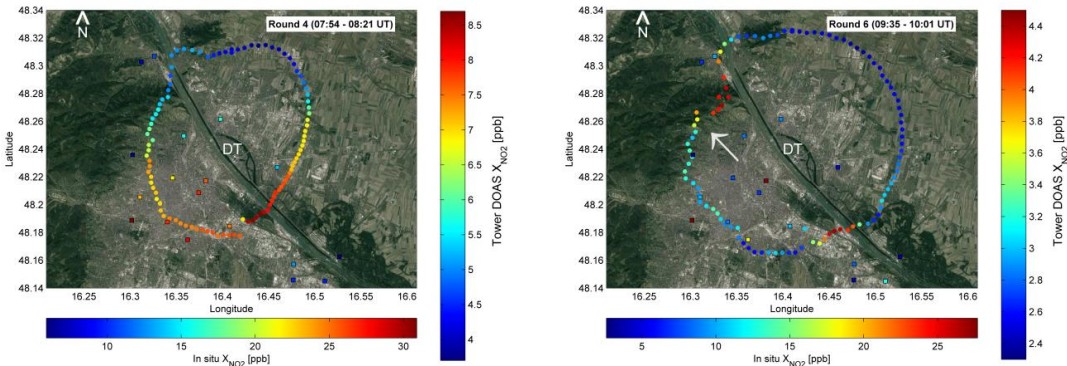

2   **Figure 18.** Spatial variability of $X_{NO2}$ in Vienna based on tower DOAS off-axis (dots) and in situ

3   surface measurements (squares) obtained on 29 April and 9 May 2016.