# Peer review of "Near-surface and path-averaged mixing ratios of NO2 derived"

_Atmospheric Chemistry and Physics, 2018_

## Referee Comment (RC1) · Anonymous Referee #1 · 10 Nov 2018

General Comments

This paper presents near-surface and path-averaged mixing ratios of NO2 derived from car DOAS zenith-sky and tower DOAS off-axis, measurements performed in Vienna city during several days on 2015 and 2016. This paper provides an useful intercomparison between tower DOAS, mobile DOAS and in-situ observations.

Specific Comments

Section 2 - Instrument and car journeys, in this section you should add few info about

the in-situ instruments (type, error, etc.). Also please add a map (a new Figure) or include in Figure 1 the location of the in-situ monitoring stations and also the location of the DOAS tower instrument.

Please describe the tower DOAS instrument, I suggest you to introduce a Table with the technical characteristics of the two instruments (tower DOAS and mobile DOAS).

Figure 2. Could you explain the very low peak of intensity? Is it related to a tree, tunnel, or a bridge? Did you filter all the DSCDs function of RMS and O4?

Figure 3, please introduce the DSCD error. Also please introduce the error of each DSCD presented in the manuscript.

3.2.1 Temporal resolution and computation of horizontal NO2 gradients- Could you specify the exposure time for the mobile DOAS instrument? (or this info could be included on the suggested Table for the two DOAS instrument).

3.2.2 Stratospheric NO2 columns, Could you specify the error of Bremen 3d CTM (B3dCTM) model?

3.2.3 Conversion to tropospheric NO2 vertical column densities SCDref, could you specify why you don't have a SCDref for each day? SCDref is quite important if you want to have qualitative data. I suggest to the authors to introduce more details about SCDref calculation, e.g. exact time of the selected SCDref. SCDref having 1.3 x 10ˆ15, 1.1 x 10ˆ15, and 2.2 x 10ˆ15 molecules/cm2 as tropospheric contribution could be realistic. Considering that SCDref contain stratospheric and tropospheric contributions, did you cancel the stratospheric contribution? why do you refer to SCDref as having only tropospheric contributions?

A chapter to describe the AMF calculation (using NO2 profiles, albedo, geometry, PBL, etc.) is mandatory for this study, I suggest to the authors to use a table. Figure 6 should be part of this section and should include the AMF calculations for several days which are presented in this study.

[Figure]

The authors should give more details about the error calculation of tropospheric NO2 VCD, or a section of errors would be more appropriate.

---

## Referee Comment (RC2) · Anonymous Referee #2 · 9 Dec 2018

General comments:

The paper presents approaches to derive near-surface and path-averaged mixing ratios from zenith-sky car DOAS and azimuth tower DOAS observations as well as a comparison with mixing ratios derived from in situ monitoring stations. Based on 9 days of car DOAS measurements and 5 days of tower measurements, acquired in 2015 and 2016, the paper provides an insight on the NO2 spatiotemporal distribution in Vienna, Austria.

[Figure]

The paper is well written and generally well-structured and provides interesting approaches to study the urban spatiotemporal NO2 distribution. The paper has improved compared to the initial submission and most comments provided in the quick review are addressed well. However, some critical issues remain and therefore my opinion has not changed that the paper would better fit in the scope of AMT than ACP.

The work has a stronger focus on the performed measurement techniques and applied retrievals approaches than on geophysical interpretation of the data, chemical/physical processes and new findings on the urban spatiotemporal NO2 distribution. I would support publication in ACP when more data and better statistics would be available in order to thoroughly assess the novel approaches and to substantiate the findings, e.g. based on long-term, routine tower DOAS and car DOAS measurements. The authors recognize the limited data set several times in the paper and foresee routine measurements based on tower DOAS off-axis and MAX-DOAS in the future.

A new, and indeed interesting, approach to convert DOAS columns to near-surface VMR (a very relevant but complex problem!) based on a linear regression analysis is introduced but not developed well in the paper. This is something that the authors recognize and attribute to the limited data/statistics available. Most of the analysis in 4.3 (comparison of car-DOAS with in-situ measurements) is not based on the new approach but on a simple assumption, assuming a constant mixing ratio in the BLH. The authors discuss that this is not necessarily valid in an urban area. I fully agree with this and I highly doubt the validity of this approach in a city, where you rather expect an exponential NO2 profile and also a strong variability over city, industry and highways. The data set is too small to fully evaluate the approach and some correlations are bad which is most likely related to the wrong assumptions in the NO2 vertical distribution. If the authors keep this approach in the paper they should at least assess the impact of other, more realistic, NO2 profiles on the statistical comparison with in situ stations and perform a sensitivity study. Eventually typical urban NO2 profiles could be derived from a high resolution CTM.

Specific comments:

P3, L9: The background signal in the reference could also be obtained by measuring one additional spectrum at 30° at the reference area and by application of the geometric approximation approach.

P10, L13: Please quantify improvement in SNR after averaging + same for averaging tower measurement on P16,L17.

P26, L3: I would elaborate a bit more on the comparison between tower VMR (at 160 m) and in-situ station VMR as this is indicated as novel in the introduction, e.g. by quantifying both instead of only giving an overall factor.

P20, L18: Please give a number on how far the air masses moved based on wind speed and time difference between the measurements. This allows to cross-check if indeed the same air masses are observed.

P24, L4: As indicated earlier, weak correlations are probably related due to wrong assumptions in the NO2 profile.

Technical corrections:

P3, L9: great advantage < added-value

P4, L9: add "for example" after estimated

P8, L12: rotations < rotation

P10, L10: drives < route

P10, L11: lines < box

P11, L26: the < an

P15, L4: In situ < in situ

P19, L5: move "are" behind "magnitude"

P19, L16: Is "temporal evolution" appropriate in the title, as you also measure spatial distribution with the moving measurement platform? Maybe split as well the car and tower measurements in different (sub)sections as they are not directly linked.

P26, L22: "unique" is not appropriate

P46 – Figure3: Please put residuals on another scale. It is not possible to check potential residual structures at this scale.
* * *

---

## Author Comment (AC1) · 14 Feb 2019

We would like to thank the reviewer for his / her useful comments.

General Comments: This paper presents near-surface and path-averaged mixing ratios of NO2 derived from car DOAS zenith-sky and tower DOAS off-axis, measurements performed in Vienna city during several days on 2015 and 2016. This paper provides an useful intercomparison between tower DOAS, mobile DOAS and in-situ observations. Specific Comments: Section 2 - Instrument and car journeys, in this section you should

add few info about the in-situ instruments (type, error, etc.). Also please add a map (a new Figure) or include in Figure 1 the location of the in-situ monitoring stations and also the location of the DOAS tower instrument. We have now rewritten the first passage of Sect. 3.3 and added information about instrument type and error of in situ instruments. In addition, we now refer to a recently published report (Spangl, 2017) (see Page 14, Line12-20). In this report, which is available online, all the air quality monitoring stations are described in detail (e.g. instrument type, location, surrounding, etc.).

We have now indicated all in situ stations in Figure 1 that are used in combination with the car DOAS measurements, but also those that are used for comparison with tower DOAS measurements. We have now renounced to include the Table with all the station name/coordinates as this information about the in situ stations as even more details can be found in the mentioned report anyway. Also now shown in Figure 1 is the position of the Danube Tower, on which tower DOAS measurements were performed. We have now included a sentence to describe what is seen in addition to the exemplary car route in this new Fig. 1 (see Page 8, Line 1-4).

Please describe the tower DOAS instrument, I suggest you to introduce a Table with the technical characteristics of the two instruments (tower DOAS and mobile DOAS).

The one and the same DOAS instrument, which we used for both car DOAS and tower DOAS applications, is already described in the version of the manuscript (see Sect. 2.1, 2.2, and 2.3 of the ACPD Manuscript Version). In order to provide more technical details of the DOAS instrument, we have now introduced a Table with the technical characteristics (see Table 1).

Figure 2. Could you explain the very low peak of intensity? Is it related to a tree, tunnel, or a bridge? Did you filter all the DSCDs function of RMS and O4?

The very low peak of intensity in Figure 2 is related to the tower DOAS measurements and shows up once every rotation of the tower platform, e.g. when the DC Tower (a

skyscraper), which is about 1 km away from the Danube Tower, blocks the field of view of the instrument. We have already described the reason for this low peak of intensity in the manuscript and used this peak for determining the exact orientation of the tower platform (see Page 8, Line 15-23 of the ACPD Manuscript Version). For the car DOAS zenith-sky measurements, we filtered all the DSCDs as a function of chisquare, e.g. NO2 DSCDs with chisquare values > 0.025 were not included in the analysis. This filtering is already described in the manuscript (see Page 9, Line 17-18 of the ACPD Manuscript Version). For the tower DOAS off-axis measurements, we did not apply any filtering. We have now added a sentence mentioning that tower DOAS off-axis measurements are not filtered (see Page 10, Line 5-7).

Figure 3, please introduce the DSCD error. Also please introduce the error of each DSCD presented in the manuscript.

We have introduced and evaluated the error for each NO2 DSCD presented in the manuscript. In general, the error of (unfiltered) DSCDs is lower than 0.75 x 1015 molec cm-2 for car DOAS zenith-sky NO2 DSCDs and lower than 1.5 x 1015 molec cm-2 for tower DOAS off-axis NO2 DSCDs presented in the manuscript (see Figures in the supplement). We have now added a sentence to give an overall (maximum) error of NO2 DSCDs for both car DOAS zenith-sky and tower DOAS off-axis measurements (see Page 10, Line 8-10).

3.2.1 Temporal resolution and computation of horizontal NO2 gradients- Could you specify the exposure time for the mobile DOAS instrument? (or this info could be included on the suggested Table for the two DOAS instrument).

Typical values of the exposure time for car DOAS zenith-sky measurements were generally between 0.00625 and 0.1 seconds. In most cases, however, the exposure time was 0.025 seconds. We have now added a sentence to specify the exposure time for the car DOAS zenith-sky measurements (see Page 10, Line 14-15 and also added this information in the new/additional Table 1.

3.2.2 Stratospheric NO2 columns, Could you specify the error of Bremen 3d CTM (B3dCTM) model?

It is difficult to quantify the accuracy of the stratospheric NO2 columns from the Bremen 3d CTM. In absolute units, it is not very good as it is a free running model without data assimilation. However, in this analysis, the stratospheric model is only used for the diurnal cycle of the stratospheric NO2 column as the absolute value is scaled to GOME2 satellite observations at the time of overpass. The uncertainty of the diurnal variation is large at twilight but small during the day as changes in stratospheric NO2 are small when compared to tropospheric NO2 columns in polluted regions. As a rough estimate, the uncertainty of the stratospheric correction is assumed to be less than 10% or typically 1 x 1015 molec cm-2. We have now added a sentence to highlight the uncertainty in the stratospheric correction (see Page 12, Line 1-6).

3.2.3 Conversion to tropospheric NO2 vertical column densities SCDref, could you specify why you don't have a SCDref for each day? SCDref is quite important if you want to have qualitative data. I suggest to the authors to introduce more details about SCDref calculation, e.g. exact time of the selected SCDref. SCDref having 1.3 x 10ËĘ15, 1.1 x 10ËĘ15, and 2.2 x 10ËĘ15 molecules/cm2 as tropospheric contribution could be realistic. Considering that SCDref contain stratospheric and tropospheric contributions, did you cancel the stratospheric contribution? why do you refer to SCDref as having only tropospheric contributions?

We agree that it is important to have as many as possible SCDref measurements for quantitative data analysis. The reason why we didn't use SCDref of each single day in our study is that for most of the days, (noontime) SCDref was taken in urban areas, where pollution levels are expected to be higher. The three SCDref measurements that we used were recorded during noontime and outside of Vienna in rather rural areas, where pollution levels are expected to be low. According to Wagner et al. (2010), AMT, spatially inhomogeneous tropospheric trace gas concentrations are a prerequisite for the "zenith-sky only" approach to avoid large systematic errors when applying Eq. 11

in their paper (Eq. 1 in our ACPD Manuscript). By using SCDref measurements in areas with rather small tropospheric NO2, as we did in our study, the errors are kept as low as possible. The selection of the three SCDref measurements was a compromise between having such measurements during noontime in unpolluted regions and at the same time to keep the time difference between SCDref and the days for which these SCDref measurements are used as low as possible (< 9 days). There are other examples in the literature where only few SCDref measurements are used for similar DOAS-type analyses (e.g. Tack et al. 2015, AMT). The authors of that study use a single SCDref measurement for a period of about 40 days. Because a similar approach to convert zenith-sky DOAS measurements into tropospheric NO2 vertical columns is described in the latter publication, we have now included this study in the references. We have specified the exact time and also the location (lat/long) of the three SCDref measurements that we used for our data analysis in Table 2. In addition, we have now added the exact time, SZA, and location of the three SCDref measurements used in our study and also we have now added more infos about how we calculated SCDref in Sec. 3.2.4 (Page 13, Line 12-21). In order to not confuse the reader, we have now avoided the use of "tropospheric" amounts in SCDref and replaced it with "residual" amounts in Sect. 3.2.4 and 3.3. This formulation was already used in the literature before (e.g. Tack et al., 2015, AMT).

A chapter to describe the AMF calculation (using NO2 profiles, albedo, geometry, PBL, etc.) is mandatory for this study, I suggest to the authors to use a table. Figure 6 should be part of this section and should include the AMF calculations for several days which are presented in this study.

In response to the comments of both reviewers, we have re-evaluated the AMFs used in the study by adding a sensitivity study of AMF changes for realistic values of AOD, single scattering albedo and mixing height in Vienna. Based on the results we have decided to change the AMF used to values based on an intermediate scenario which according to our sensitivity study provides a good compromise. All other scenarios are

well within 20% of these values. Therefore, we have re-calculated VCDtropo NO2 in our study and plotted the respective figures again. Both the sensitivity study and the description of the AMF used have been included in a new Section (Sect. 3.2.3, Page 12, Line 8-27) in the revised manuscript.

The authors should give more details about the error calculation of tropospheric NO2 VCD, or a section of errors would be more appropriate.

Uncertainties in tropospheric VCDs are introduced by uncertainties in the quantities used in equation 1 of the manuscript. Assuming that the stratospheric AMF is well known, the uncertainties of DSCDmeas, SCDref, SCDstrato and AMFtropo need to be considered: (see Equation in the supplement) an overall uncertainty of 25% is found, dominated by the assumed 20% uncertainty of the AMF. For situations approaching twilight, the absolute uncertainty of the stratospheric correction increases, and the relative uncertainty of the slant column can become the dominating error source. If the background measurement SCDref cannot be taken in a clean region, then the absolute uncertainty on this quantity can become large and important for the overall uncertainty (see Wagner et al., 2010). We have now added a paragraph in order to provide information on the overall uncertainty of VCDtropo in our study (see Page 13, Line 27-28; Page 14, Line 1-9).

---

## Author Comment (AC2) · 14 Feb 2019

We would like to thank the reviewer for his / her useful comments.

General comments: The paper presents approaches to derive near-surface and path-averaged mixing ratios from zenith-sky car DOAS and azimuth tower DOAS observations as well as a comparison with mixing ratios derived from in situ monitoring stations. Based on 9 days of car DOAS measurements and 5 days of tower measurements, acquired in 2015 and 2016, the paper provides an insight on the NO2 spatiotemporal

distribution in Vienna, Austria. The paper is well written and generally well-structured and provides interesting approaches to study the urban spatiotemporal NO2 distribution. The paper has improved compared to the initial submission and most comments provided in the quick review are addressed well. However, some critical issues remain and therefore my opinion has not changed that the paper would better fit in the scope of AMT than ACP. The work has a stronger focus on the performed measurement techniques and applied retrievals approaches than on geophysical interpretation of the data, chemical/physical processes and new findings on the urban spatiotemporal NO2 distribution. I would support publication in ACP when more data and better statistics would be available in order to thoroughly assess the novel approaches and to substantiate the findings, e.g. based on long-term, routine tower DOAS and car DOAS measurements. The authors recognize the limited data set several times in the paper and foresee routine measurements based on tower DOAS off-axis and MAX-DOAS in the future.

We agree that our manuscript might also fit in the scope of AMT. We do not fully agree with the argument that more data are needed to publish such a study in ACP. On the one hand, we have clearly defined our study as a "case study". On the other hand, there are other studies in ACP which evaluate data from only few days. "Estimation of NOx emissions from Delhi using Car MAX-DOAS observations and comparison with OMI satellite data" by Shaignafar et al. (2011), a well cited ACP study, is only one example.

A new, and indeed interesting, approach to convert DOAS columns to near-surface VMR (a very relevant but complex problem!) based on a linear regression analysis is introduced but not developed well in the paper. This is something that the authors recognize and attribute to the limited data/statistics available. Most of the analysis in 4.3 (comparison of car-DOAS with in-situ measurements) is not based on the new approach but on a simple assumption, assuming a constant mixing ratio in the BLH. The authors discuss that this is not necessarily valid in an urban area. I fully agree with

this and I highly doubt the validity of this approach in a city, where you rather expect an exponential NO2 profile and also a strong variability over city, industry and highways. The data set is too small to fully evaluate the approach and some correlations are bad which is most likely related to the wrong assumptions in the NO2 vertical distribution. If the authors keep this approach in the paper they should at least assess the impact of other, more realistic, NO2 profiles on the statistical comparison with in situ stations and perform a sensitivity study. Eventually typical urban NO2 profiles could be derived from a high resolution CTM.

We agree that the newly introduced linear regression analysis is not yet developed well, which mainly depends on the availability of data for only few days. Nevertheless, we argue that it is meaningful to present a new method to convert VCDtropo into near-surface mixing ratios, even if only limited data is available. The collection of data, which is available for testing this new method, was well thought out and a lot of effort and time was spent to get this unique data set. There is no comparable study, which collected data for one and the same car route for many times as well as for many different meteorological conditions. We argue that our data, collected in an urban environment, in combination with a relatively large number of air quality monitoring stations, is exactly what we need for introducing and testing such a new method. We agree that most of the analysis in Sect. 4.3 is based on the method of Knepp et al. (2013). From this analysis we found that for some days (mostly when air masses came from southeastern directions and when wind speeds were rather low) the correlation was high but slope and intercept were not satisfying enough, most probably because of the fact that the assumption of a constant mixing ratio within the PBL does not work for urban environments having different meteorological conditions. Again, our intention was to perform such car DOAS zenith-sky measurements on days with different meteorological conditions to see how these changing conditions affect the assumption of a constant mixing ratio within the PBL. The findings of this analysis raised the motivation to go one step further and test a new method – a method that seems to reduce the complexity of the problem of converting DOAS columns to near-surface mixing ratios, without deriving

typical NO2 profiles from highly resolved CTM, which also have well known problems in representing the vertical distribution of NO2 in complex urban environments. Our aim was not to fully evaluate this method but rather introduce it and test it on a unique data set. We agree that deriving typical urban profiles from high resolution CTMs and perform sensitivity analysis to assess the impact of more realistic NO2 profiles is an interesting and worthwhile suggestion. However, the main motivation of this work was to evaluate a new method, and as shown in Fig. 17, this method appears to perform very well for at least for our data set.

Specific comments: P3, L9: The background signal in the reference could also be obtained by measuring one additional spectrum at 30° at the reference area and by application of the geometric approximation approach.

We agree that additional measurements at EA = 30° would help in this case. Unfortunately, such measurements were not performed and are thus not available. Nevertheless, we will consider such measurements for future car DOAS measurements.

P10, L13: Please quantify improvement in SNR after averaging + same for averaging tower measurement on P16, L17. I t is not clear to us what the reviewer would like to see here. Averaging reduces the variability in NO2 signal as expected, and this is illustrated in Figure 4 in the manuscript. Raw data (0.05 seconds) appear to have a random scatter of the order of 8 x 1015 molec cm-2 peak-to-peak, which is reduced to less than 1 x 1015 molec cm-2 in the averaged data (5 seconds). Thus one could say that the signal to nose ratio has improved by a factor of 8. However, as also seen in Fig. 4, it is not trivial to distinguish between measurement related noise and real atmospheric variability, and thus it is in our opinion not clear what the real improvement in SNR is.

P26, L3: I would elaborate a bit more on the comparison between tower VMR (at 160 m) and in-situ station VMR as this is indicated as novel in the introduction, e.g. by quantifying both instead of only giving an overall factor.

Due to the fact that data is only available for a couple of days, and reasonable comparison between tower and in situ NO2 mixing ratios can only be made for the two rotations of 29 April and 9 May 2016, quantification is challenging. Nevertheless, we have now added a new figure (Fig. 21) to compare the NO2 mixing ratios derived from tower DOAS off-axis measurements with the one calculated from surface NO2 concentrations. The comparison is based on round 4 and round 6 of 29 April and 9 May 2016, respectively (e.g. the same two rounds as presented in Fig. 18, Fig. 19, and Fig. 20). We have computed the mean and standard deviation of tower DOAS off-axis NO2 mixing ratios of the full tower rotation and the mean and standard deviation of in situ NO2 mixing ratios from those stations which are within the circle as determined by hOPL. The results are described in Sect. 4.5 (Page 29, Line 1-18) and Sect. 5 (Page 31, Line 23-24) and highlighted in the abstract.

P20, L18: Please give a number on how far the air masses moved based on wind speed and time difference between the measurements. This allows to cross-check if indeed the same air masses are observed.

When considering round-averaged wind directions, wind speeds and 1.5 hours for the time difference between the measurements at one and the same location, air masses on 10 April 2015 moved about 5.85 km (from the first to the second round) and 8.1 km (from the second to the third round). Consequently, in total those air masses moved about 14 km, which is in good agreement with the position of the NO2 peak of round 1 (red) at about 20 km and the position of the second of the two NO2 peaks of round 3 (blue) at about 6 km (see Fig. 11). We note that the 3-rounds averaged wind direction of that day (125.3 deg) slightly differs from the position of the A22 highway (~150 deg), which was considered for this case study.

P24, L4: As indicated earlier, weak correlations are probably related due to wrong assumptions in the NO2 profile.

We agree that weak correlations are probably related due to wrong assumptions in

the NO2 profile, in addition to changing air masses with sometimes only low pollution levels. As argued above, we conclude that using the method of Knepp et al. (2013) assuming constant mixing of NO2 within the PBL does not work as good for all days of our study performed in the urban environment of Vienna. This fact was basically the motivation to test a new method, e.g. the linear regression analysis, which also accounts for other meteorological parameters that could have an effect on NO2 profiles, e.g. wind speed. Due to the good correlation between modeled and measured NO2 surface mixing ratios (R = 0.94) achieved with this new introduced and tested method we can argue that NO2 profiles are not essentially needed for the conversion of VCDtropo into mixing ratios as wind speed, na, MH seem to strongly affect NO2 profiles, at least over the urban area of Vienna, and at least for the data we have analyzed. This is generally the main message of our introduced and tested method. In the future, we will apply this method to zenith-sky measurements from operating MAX-DOAS instruments in Vienna, where better statistics are available. While weak correlation is found when using the method of Knepp et al. (2013), a very high correlation is found with our new method. This makes it worthwhile enough to publish this method and to motivate other research group to work on this complex problem of converting DOAS columns to surface mixing ratios.

Technical corrections: P3, L9: great advantage < added-value P4, L9: add "for example" after estimated P8, L12: rotations < rotation P10, L10: drives < route P10, L11: lines < box P11, L26: the < an P15, L4: In situ < in situ P19, L5: move "are" behind "magnitude"

We have considered all the "technical corrections" in the new version of the manuscript.

P19, L16: Is "temporal evolution" appropriate in the title, as you also measure spatial distribution with the moving measurement platform? Maybe split as well the car and tower measurements in different (sub)sections as they are not directly linked.

We agree that "temporal evolution" is not meaningful enough in this case and thus,

changed it into "spatio-temporal patterns" (see Page 21, Line 22-23). We also agree that for a better overview, splitting car and tower DOAS (Sect. 4.2) is the right way. We have now added a new (sub)section (Sect. 4.3) (see Page 23, Line 21-22). We have now also added "obtained from tower DOAS off-axis" in the title of Sect. 4.5 (see Page 27, Line 12).

P26, L22: "unique" is not appropriate

We have now removed "unique" in the first sentence of the summary and outlook sections.

P46 – Figure3: Please put residuals on another scale. It is not possible to check potential residual structures at this scale

We have now put residuals on a different scale to make them more readable (see Page 51).

---

## Author Response (AR2)

**Author's response to comments by referee #1**

We would like to thank the reviewer for his / her useful comments.

*General comments:*

*The authors present a study that combines observations from a mobile DOAS system and a stationary system that rotates through every azimuth approximately hourly. Additionally, they pull in data from in situ samplers to compare the column-converted NO2 mixing ratio to in situ observations. In my view the novelty of this experiment is the combination of observations, with little or no novelty in the presented chemistry/physics, making it by far more applicable to AMT as opposed to ACP. While this paper could be considerably shorter (by focusing on the technique and omitting unnecessary discussion of chemistry that is well understood), I recommend it for publication after some minor revisions.*

*Specific Comments:*

*page 8 line 10: "anti-clockwise" should be changed to counter-clockwise*

We have now changed "anti-clockwise" into "counter-clockwise".

*page 8 line 28 - page 9 line 2: there is no need to "assume" the rotation speed. You can calculate this from your observations.*

We have now omitted "assuming" and rephrased the sentence accordingly.

*page 20 line 22: it will be helpful to the reader if you identify the day of week corresponding to each date.*

We have now added the day of week in parenthesis after each of the three dates.

*page 20 line 24-26: Does "starting point" reference the starting point of Fig. 1? This says that "as we go farther from the clean area NO2 increases". Would it make more sense to reference the distance from the center of the city, which would better show the gradient from the "source" to background?*

"Starting point" does not reference the starting point of Fig. 1. "Starting point" is the start point of individual car DOAS zenith-sky measurements at the full resolution of 0.05 seconds (see page 11, line 3-11).

*page 21 line 19: What do you mean by "horizontal NO2 scale"? Is this the e-folding distance?*

We agree that this is a somewhat qualitative statement. The idea is that over this distance, the NO2-field is more homogeneous than over longer distances, implying that this is a typical size of NO2 plumes observed in the columns. This is of course related to the e-folding distance but given the complex mix of pattern of emission sources, transport, and mixing and photochemistry of the plume, a single e-folding distance has little meaning within an urban area.

*page 22 lin 16: Please clarify what these starting/ending points are (same as Fig. 1?)*

Also here the starting and end points are not referenced to Fig. 1. Here, starting and end points define the distance of the A22 motorway for evaluating the NO2 variation along this section of an individual car DOAS route.

*page 23 line 4: replace "left" and "right" with "east" and "west" as necessary.*

We have now changed it accordingly.

*page 25 line 22: I see nothing on 2-October near the 1:1 line. Please clarify.*

We agree that it is not close to 1:1 line. We have now rephrased the sentence accordingly.

*page 27 line 21: define "hOPL" after first use.*

hOPL is defined after first use on page 20, line 1.

*fig. 11: is the distance here the same as in Fig. 10?*

The distance is not the same. The time series in Fig. 10 present measurements of the whole day (e.g. measurements of three circuits as shown in Fig. 1 plus an additional circle in an rural area outside of Vienna), while time series of Fig. 11 present the selected section of an individual round (A22 motorway, along the Danube River) as described in Sect. 4.2.

*fig. 18: please define hOPL*

We have now defined hOPL in the figure caption of Fig. 18.

**Author's response to comments by referee #2**

We would like to thank the reviewer for his / her useful comments.

*General comments:*

*Is still unclear to me if the SCDref (in the author's opinion ) contain tropospheric and stratospheric absorptions. The authors present that the SCDref calculation is based on ground-based tropospheric in-situ measurements. Please confirm if the stratospheric contribution to the SCDref is neglected or canceled.*

SCDref contains both tropospheric and stratospheric absorptions. Stratospheric NO2 amounts are calculated from B3dCTM simulations and tropospheric residual amounts of SCDref are calculated by assuming the empirical relationship between VCDtropo and in situ NO2 mixing ratios as reported in Kramer et al. (2008).